# The molecular basis of Human FN3K mediated phosphorylation of glycated substrates

Ankur Garg [1,2,6], Kin Fan On [1,2,6], Yang Xiao [3,4], Elad Elkayam[1,5], Paolo Cifani[1], Yael David [3,4] & Leemor Joshua-Tor [1,2] ✉

Glycation, a non-enzymatic post-translational modification occurring on proteins, can be actively reversed via site-specific phosphorylation of the fructose-lysine moiety by FN3K kinase, to impact the cellular function of the target protein. A regulatory axis between FN3K and glycated protein targets has been associated with conditions like diabetes and cancer. However, the molecular basis of this relationship has not been explored so far. Here, we determined a series of crystal structures of HsFN3K in the apo-state, and in complex with different nucleotide analogs together with a sugar substrate mimic to reveal the features important for its kinase activity and substrate recognition. Additionally, the dynamics in sugar substrate binding during the kinase catalytic cycle provide important mechanistic insights into HsFN3K function. Our structural work provides the molecular basis for rational small molecule design targeting FN3K.

Protein glycation is a posttranslational modification involving the nonenzymatic attachment of reducing sugars, like glucose and ribose, onto the free amino groups of basic protein residues (lysine and arginine) via a Maillard reaction, leading to the formation of Amadori products[1]. This reaction begins with the spontaneous attachment of a sugar moiety to the amino group of the basic residue to form a Schiff base, which then slowly rearranges to form a ketosamine[2,3]. Glycated protein residues are more reactive and could crosslink to the extracellular matrix, and could also generate superoxide radicals due to their highly reducing nature[4,5]. This process has been implicated in multiple chronic diseases, including arthritis, atherosclerosis, and diabetes[6–9]. To keep this deleterious effect in check in vivo, enzymes exist to counteract protein glycation by metabolizing these Amadori products. For instance, some fungi and bacteria use fructosyl amino acid oxidases (also called Amadoriases) to metabolize Amadori products via a Schiff base intermediate to revert the glycation modification[10–13]. Interestingly, a crystal structure of a fungal Amadoriase revealed that the active site is located in a deep crevice,

suggesting that substrates that have a fructosamine moiety on a long side chain, for example, that is connected to a protein polypeptide chain, may have better accessibility to the catalytic site[14]. Alternatively, the sugar moiety on glycated residues can be phosphorylated by a class of small-molecule kinases called fructosamine-3-kinases (FN3Ks), which are present in mammals, birds, and plants[15,16].

FN3K, initially isolated from human erythrocytes[17,18], was shown to be specific for a 1-deoxy-1-amino fructose adduct, and could tolerate a bulky group at the N1 position of a fructose-containing substrate[18]. Further characterization indicated that FN3K is responsible for protein deglycation by phosphorylating protein-associated fructosamines[18–20]. It was proposed that phosphorylation on the O3' of the sugar moiety results in an unstable species that deprotonates C1 to facilitate the phosphate removal and generate the Schiff base intermediate. The Schiff base intermediate is further hydrolyzed to separate the sugar from the amino group as a 2-keto-3-deoxyaldose[15]. Thus, the ketoaldehyde generated is converted to the harmless 3-deoxy-2-ketogluconic acid (DGA)[21–26], while the protein residue returns to a

[1]Cold Spring Harbor Laboratory, One Bungtown Road, Cold Spring Harbor, New York 11724, USA. [2]Howard Hughes Medical Institute, Cold Spring Harbor Laboratory, One Bungtown Road, Cold Spring Harbor, New York 11724, USA. [3]Chemical Biology Program, Memorial Sloan Kettering Cancer Center, New York, NY, USA. [4]Tri-Institutional PhD Program in Chemical Biology, New York, NY, USA. [5]Present address: Ventus Therapeutics, Waltham, Massachusetts 02453, USA. [6]These authors contributed equally: Ankur Garg, Kin Fan On. ✉e-mail: leemor@cshl.edu

non-glycated state, highlighting FN3K's role in repairing glycated proteins[15,20].

Perhaps one of the most important examples of glycation-mediated control is the critical transcription factor NRF2. NRF2 is a cap'n'collar (CNC) basic leucine zipper (bZIP) transcription factor that regulates the expression of more than 200 genes in several cellular pathways, including redox balance, metabolic reprogramming, energy production, and biomolecule syntheses, and also confers a growth advantage to transformed cells in tumorigenesis[27,28].

A recent study uncovered the link between NRF2 glycation and FN3K dependency in cancer and identified FN3K as a potent NRF2 activator in malignancies[29]. Whereas NRF2 glycation (of K462, K472, K487, R499, R569, R587) was shown to affect both its stability and transactivation function, FN3K reversed these effects likely by deglycation[29]. Downregulation of FN3K in the liver (HepG2, Huh1 cell lines) and lung (H3255, H460 cell lines) cancer cell lines resulted in the impairment of NRF2 function by reducing its protein stability and disrupting its dimerization with the small musculoaponeurotic fibrosarcoma (sMAF) proteins, all of which are critical for NRF2 function[29–31]. Further analysis showed that FN3K knockdown resensitized human NSCLC cell lines (H3255 and PC9) to erlotinib treatment[29], highlighting the therapeutic potential of targeting FN3K in cancer cells that exhibit survival dependency on NRF2. It is currently unclear what mechanistic role FN3K plays in modulating NRF2 function in cancer. However, similar to FN3K's roles in the pathology of other diseases like diabetes, its phosphorylation activity is believed to be crucial for its regulation of NRF2.

In this work, to explore FN3K therapeutic potential and to further understand the FN3K-mediated glycation control on NRF2, we report a series of crystal structures of human (Hs) FN3K in the apo, and in substrate-bound forms with various nucleotides representing different catalytic states. Our mutational analysis further validates the structural findings to provide valuable functional insights, based on the structure-activity relationships (SAR), for small-molecule inhibitor design against FN3K in regulating the FN3K-NRF2 axis.

## Results

### HsFN3K is a deglycase in vitro

While NRF2 was thought to be an FN3K substrate in vivo[29], there is no direct evidence for the removal of sugar adducts on NRF2 by FN3K. To establish NRF2 deglycation by FN3K, we designed a peptide (H-LALIKDIQ) spanning the C-terminal region of HsNRF2 (aa 495-501) that was shown to be glycated by trypsin mis-cleavage[29]. To improve the sensitivity and dynamic range of our assays, we replaced the arginine residue in the native sequence (R499) with a lysine, as lysine is more reactive towards reducing sugars than arginine[32] and forms a ketosamine, the substrate of FN3K, when reacted with an aldose[3]. We also chose D-ribose as the glycating agent due to its higher reactivity than other sugars[33], and a shorter half-life ( ~ 25 min) of the generated ribulosamine 3-phosphate compared to other ketosamine 3-phosphates like fructosamine 3-phosphate (half-life of ~7 h)[34] in order to facilitate demonstration of FN3K-mediated deglycation (Fig. 1A). Using ultra-performance liquid chromatography coupled with mass spectrometry (UPLC-MS), we successfully detected mass adducts corresponding to the Schiff base ($[M + 132]_A$) and Amadori product ($[M + 132]_B$) on the peptide treated with D-ribose (Fig. 1C). Although both adducts are of the same mass, we were able to distinguish them by the diagnostic fragmentation pattern of the Schiff base (Supplementary Fig. 1) in accordance with those reported in the literature[35]. Next, to test FN3K-mediated deglycation, we incubated the glycated peptide with the purified full-length (FL) HsFN3K (Fig. 1B) in the presence or absence of ATP. Gratifyingly, we observed ATP-dependent deglycation of the Amadori product by FN3K (Fig. 1D). Moreover, we were able to detect the relatively unstable ribulosamine 3-phosphate intermediate by its unique mass shift of

+212, demonstrating the mechanism of FN3K-mediated deglycation through the direct phosphorylation of the Amadori product (Fig. 1E).

Due to the heterogeneity of the glycated adducts on NRF2 peptides, we turned to a small molecule substrate 1-deoxy-1-morpholino-D-fructose (DMF) for our subsequent studies. DMF contains a 6-carbon sugar moiety linked to the morpholino group via a nitrogen atom, which mimics a glycated tail attached to the nitrogen of basic amino acids (lysine and arginine). The FL-HsFN3K purified from either insect cells or *E. coli* cells both phosphorylated DMF with similar high kinase activity (Fig. 1F). We also observed a dimeric FN3K species during the purification from insect cells (Supplementary Fig. 2A-B), which exhibited ~60% higher kinase activity on DMF as compared to the monomeric FN3K species (Fig. 1F). Dimeric FN3K exhibits a similar thermal melting spectrum with a melting temperature ($T_m$) of 54 °C compared to 54.5 °C for monomeric HsFN3K (Supplementary Fig. 2C). The dimeric FN3K's sensitivity to different protease enzymes is also quite similar to monomeric HsFN3K (Supplementary Fig. 2D), suggesting that the dimeric HsFN3K has a similar structural fold as the monomeric species, and the increased kinase activity of the HsFN3K dimer might be due to higher substrate or nucleotide turnover. The purified dimeric FN3K partially re-equilibrates to the monomeric FN3K under non-reducing conditions, while it gets converted to a monomer under reducing conditions in 20 min (Supplementary Fig. 2F–G), suggesting that HsFN3K dimerization is redox-state dependent, as has been observed for AtFN3K[16]. Taken together, our in vitro biochemical characterization confirms that HsFN3K is an active kinase, capable of phosphorylating small molecules, and serves as mechanistic support for the NRF2 glycation and FN3K-mediated deglycation cycle.

### Overall architecture of the HsFN3K

To investigate the structural basis of FN3K kinase activity, we determined a crystal structure of HsFN3K in its apo state. Despite extensive screening, FL-HsFN3K was recalcitrant to crystallization. We, therefore, designed an internal loop truncated HsFN3K (HsFN3KΔ) guided by limited proteolysis experiments (Supplementary Fig. 2D, E). HsFN3KΔ is also catalytically active and phosphorylates DMF substrate over time, however with about 60% activity compared to WT HsFN3K (Fig. 1F). HsFN3KΔ crystallized in space group P2₁2₁2₁ with two molecules in the asymmetric unit (ASU) (Table 1). The crystal structure was determined by molecular replacement using PHASER with the C-lobe of AtFN3K[16] as a search model, while the N-lobe was manually built into the electron density map. The whole HsFN3KΔ polypeptide could be unambiguously traced in the 1.67 Å resolution electron density map, revealing a globular bi-lobal structure with the N-lobe and C-lobe exhibiting a canonical protein-kinase-like (PKL)-fold. The two molecules in the ASU are almost identical, with an RMSD of 0.21 Å over 201 Cα atoms. The two C-lobes in the ASU are placed in a head-to-head orientation with the N-lobe α1 helix from one protomer positioned onto the N-lobe β-sheet of the second, forming a domain-swapped dimer (Fig. 1G). The dimeric arrangement of the two HsFN3K protomers is supported by an intermolecular di-sulfide linkage between the side chain of a Cys24 residue from the ATP-binding P-loop of each protomer (Fig. 1H). A C24S mutation in HsFN3K results in an exclusively monomeric FN3K species based on size-exclusion chromatography and non-reducing SDS-PAGE analysis (Supplementary Fig. 2F, G), confirming that the redox-dependent HsFN3K dimeric form is Cys24 dependent. This cysteine residue is highly conserved in FN3K among different species. The other eukaryotic FN3K structure from *Arabidopsis thaliana* (At) also exhibits a cysteine-mediated dimerization[16] (Supplementary Fig. 3C, D). However, in contrast to the AtFN3K dimer, which is either catalytically inactive or much less active[16], the HsFN3K dimer exhibited significantly higher kinase activity compared to its monomeric species. Though we can not rule out that some HsFN3K dimer gets converted to monomers during these assays, an increased kinase activity is suggestive of the presence of

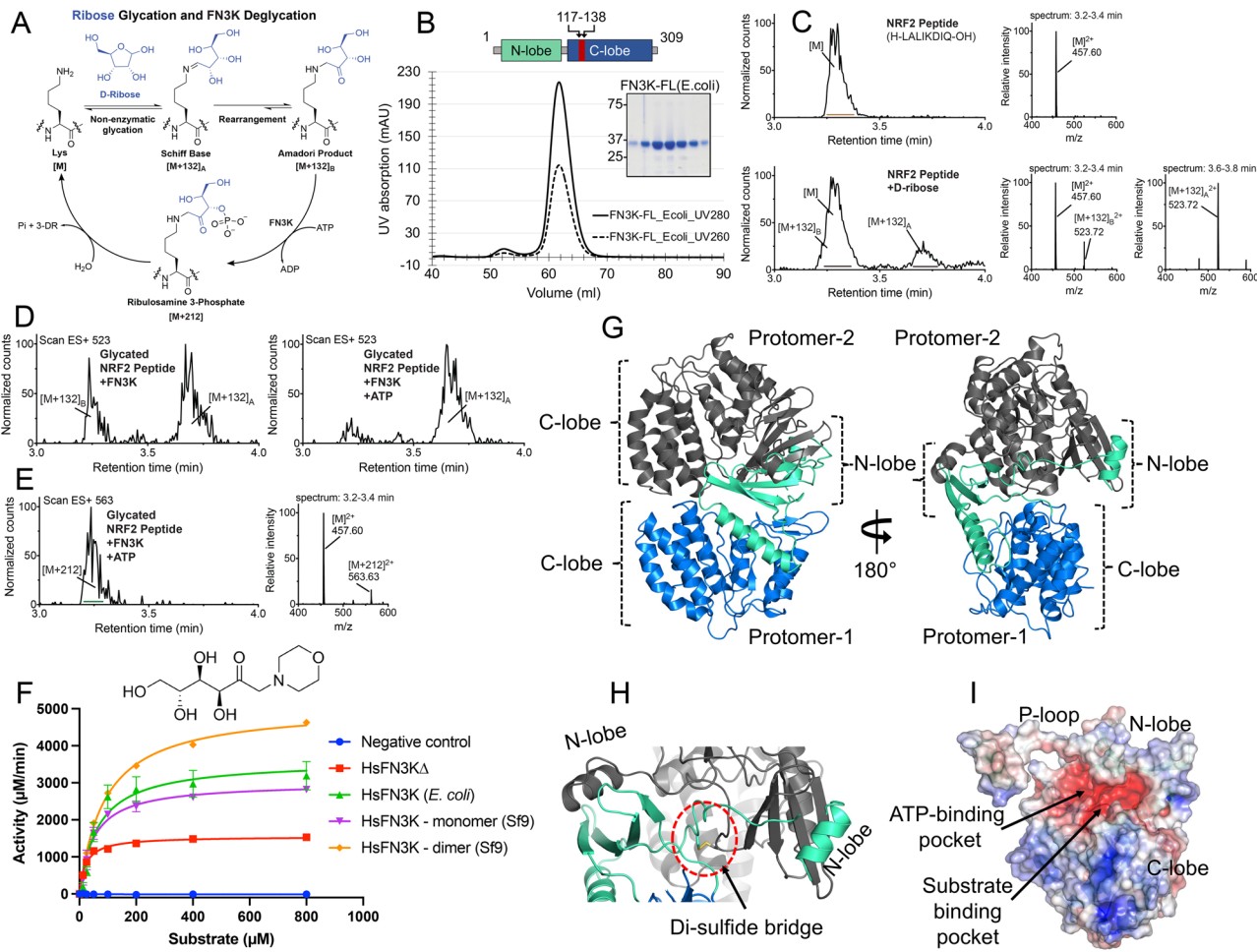

**Fig. 1 | HsFN3K is an active kinase. A** Mechanism of D-ribose glycation and FN3K-mediated deglycation of a lysine substrate. **B** Size-exclusion chromatogram (SEC) and SDS-PAGE showing the homogeneously purified full-length (FL) HsFN3K protein from E. coli. The UV280 and UV260 traces from a representative SEC run are shown in solid and dotted lines respectively ($n = 3$). Domain architecture showing the N-lobe (cyan) and C-lobe (blue) of HsFN3K is also shown. The truncated loop (aa 117-138) from the C-lobe to generate HsFN3KΔ is colored red (**C**) Synthetic NRF2 peptide (H-LALIKDIQ-OH, M.W. = 912.56 g/mol) was incubated with excess D-ribose in PBS for 24 h at 37 °C and analyzed by UPLC-MS. Representative chromatograms and combined mass spectra for the unglycated (top) and glycated (bottom) peptides were shown. **D** Extracted ion chromatograms of m/z 523 ([M + 132]²⁺) reveal that FN3K deglycates the Amadori product ([M + 132]$_B^{2+}$) in an ATP-dependent manner. **E** Extracted ion chromatograms of m/z 563 and the combined mass

spectrum confirm the presence of the phosphorylated intermediate, ribulosamines 3-phosphate ([M + 212]), following FN3K treatment with ATP. **F** HsFN3K in vitro kinase assays on DMF substrate. All the HsFN3K proteins showed kinase activity. Data are presented as mean ± SD ($n = 3$). **G** Cartoon representation of the crystal structure of apo-HsFN3K showing two molecules arranged as a domain-swapped dimer via the N-lobe. The domains in protomer-1 are colored as shown in the domain architecture, while protomer-2 is shown in grey (**H**) Shown is the Cys24-mediated disulfide bridge, supporting the domain-swapped dimeric arrangement. **I** Electrostatic surface potential of the apo-HsFN3K showing overall negatively charged pockets for ATP and substrate binding. The electrostatic surface potential is displayed in a range from -5 (red) to +5 kT/e (blue). Source data are provided as a Source Data file.

FN3K dimers. These observations indicate that cysteine-mediated dimerization could serve as a regulatory feature of this enzyme.

The electrostatic surface of the Apo-HsFN3K structure exhibits an overall negatively charged pocket, decorated with polar residues, near the P-loop for ATP binding, and a sulfate ion is observed in one protomer occupying the space for the nucleotide β-phosphate in this structure. Adjacent to it, the substrate binding pocket is also negatively charged and is decorated with polar residues (Fig. 1I).

HsFN3K and AtFN3K have ~ 35% sequence identity, and their structural superposition revealed that the overall PKL-fold is conserved in both eukaryotic FN3Ks with an RMSD of 1.3 Å. Notably, in contrast to a fully extended conformation of the P-loop in AtFN3K, the P-loop in HsFN3K packs against the N-lobe β-sheet. This P-loop reorients the N-lobe α1-helix at a ~ 66° angle apart in the two structures (Supplementary Fig. 3E), facilitating a distinct dimeric assembly in the HsFN3K crystal structure. Furthermore, a Dali protein structure

comparison[36] revealed that HsFN3K shares structural similarity with a putative FN3K from bacteria (PDBid 3JR1) (Supplementary Fig. 3F) with an RMSD of 2.0 Å, further underscoring that HsFN3K has a typical kinase fold.

**Glycated substrate recognition by HsFN3K**

To understand how FN3K binds glycated substrates, we determined crystal structures of HsFN3KΔ in complex with ADP and a small molecule glycated substrate mimic, DMF. The structure was determined by MR using PHASER and the apo-HsFN3KΔ structure as a search model (Table 1). The protein fold is almost identical to the apo-HsFN3KΔ structure (Fig. 2A) and exhibits unambiguous electron density for ADP and DMF in the nucleotide- and substrate-binding sites, respectively (Supplementary Fig. 4A). The portion of the P-loop preceding the disulfide-bridged cysteines is unstructured, as no electron density is observed for it (Supplementary Fig. 4A). Kinases have a

**Table-1 | Summary of X-ray data collection and structure refinement statistics for different crystal structures of HsFN3K**

| | Apo | AMPPNP-DMF | ADP-DMF (I) | ATP-DMF | ADP-DMF (II) | FN3K (D217S) ATP |
|---|---|---|---|---|---|---|
| PDBid | 9CX8 | 9CXN | 9CXV | 9CXM | 9CXW | 9CXO |
| Beamline | ID-24E | ID-24E | AMX-17ID-1 | ID24-E | ID24-E | ID24-E |
| Wavelength (Å) | 0.979180 | 0.979180 | 0.92011 | 0.979180 | 0.979180 | 0.979180 |
| Data collection on | 111920 | 111920 | 030421 | 111021 | 111021 | 030422 |
| Space group | $P2_12_12_1$ | $P2_12_12_1$ | $P2_12_12_1$ | $P2_12_12_1$ | $P2_12_12_1$ | $P2_12_12_1$ |
| Cell dimensions | | | | | | |
| a, b, c (Å) | 52.35, 111.53, 132.60 | 52.60, 111.52, 132.91 | 52.80, 112.00, 132.90 | 52.93, 112.28, 132.80 | 52.85, 111.76, 132.66 | 52.31, 112.45, 132.93 |
| Resolution (Å) | 132.60–1.67 (1.70–1.67)$^a$ | 132.9–1.90 (1.94–1.90) | 29.72–1.80 (1.85–1.80) | 132.80–1.76 (1.87–1.76) | 132.66–1.80 (1.91–180) | 132.9–2.32 (2.44–2.32) |
| No of reflections | 295,124 | 231,201 | 503,659 | 441,699 | 553,663 | 156,265 |
| Unique reflections | 90,249 | 62,294 | 73,993 | 79,306 | 73,858 | 34,801 |
| $R_{merge}$ (%) | 4.7 (73.9) | 4.3 (29.9) | 5.6 (63.7) | 4.3 (47.9) | 5.8 (110) | 9.7 (67.7) |
| $<I/\sigma(I)>$ | 15.8 (1.6) | 18.5 (3.6) | 18.00 (2.40) | 23.2 (3.1) | 19.2 (1.6) | 12.0 (2.2) |
| $CC_{1/2}$ | 0.99 (0.57) | 0.99 (0.93) | 0.99 (0.85) | 0.99 (0.89) | 99.9 (75.7) | 0.99 (0.73) |
| Completeness (%) | 99.1 (96.0) | 99.6 (98.8) | 99.9 (98.3) | 99.8 (98.2) | 99.8 (96.4) | 99.3 (96.3) |
| Multiplicity | 3.3 | 3.8 | 6.8 | 5.6 | 7.5 | 4.5 |
| Refinement | | | | | | |
| $R_{work}/R_{free}$ | 0.174/0.205 | 0.176/0.221 | 0.177/0.205 | 0.177/0.208 | 0.187/0.220 | 0.200/0.237 |
| No of atoms | | | | | | |
| Protein | 4616 | 4551 | 4568 | 4579 | 4589 | 4597 |
| Hetero atoms | 119 | 151 | 152 | 150 | 116 | 129 |
| Solvent atoms | 593 | 537 | 527 | 534 | 388 | 173 |
| RMSD | | | | | | |
| Bond lengths (Å) | 0.017 | 0.020 | 0.018 | 0.017 | 0.014 | 0.002 |
| Bond angles (°) | 1.455 | 1.577 | 1.559 | 1.35 | 1.30 | 0.565 |
| Ramachandran statistics | | | | | | |
| Most favored (%) | 97.6 | 97.7 | 96.8 | 96.8 | 97.0 | 97.4 |
| Allowed (%) | 2.4 | 2.3 | 3.2 | 3.2 | 2.9 | 2.4 |
| Outliers (%) | 0.1 | 0.0 | 0.3 | 0.0 | 0.0 | 0.2 |
| Rotamer outlier (%) | 0.5 | 0.0 | 0.22 | 0.2 | 0.2 | 0.6 |

$^a$Values in parentheses represent the highest resolution shell.

universally conserved loop at the core of their C-lobe, known as the "catalytic loop" which supports both nucleotide and substrate binding. Similarly, in HsFN3K, the catalytic loop (aa 215-224) is positioned very close to the ADP and DMF, stabilizing both of them (Fig. 2B).

The ADP base fits in the negatively charged P-site adjacent to the β-sheet of the N-lobe. Several bulky amino acids (Pro71, Met88, His90, Met93, Phe39, and Tyr214) form a narrow pocket where the adenine base fits, stacked between the side chains of Phe39 and Met93. The main chain carbonyl oxygen of Glu89 and the main chain amide of Leu91 recognize the adenine via direct hydrogen (H)-bonds with N6 and N1 atoms, respectively (Fig. 2B). Notably, this loop containing Leu91 and Glu89 shows subtle outward movement to accommodate the adenine base, compared to the apo-HsFN3K structure (Supplementary Fig. 4B). The two phosphates coordinate the $Mg^{2+}$ ion in the catalytic site and are recognized by the highly conserved Lys41 residue (Fig. 2B, Supplementary Fig. 4A). This interaction is known to be critical for the activity of eukaryotic protein kinases (ePKs)[37], since it positions the nucleotide in a catalytically-competent geometry. The HsFN3K Asp234 directly interacts with the $Mg^{2+}$ ion and is also considered to be a catalytic residue. A mutation of Asp234 to an asparagine completely abolishes the kinase activity of HsFN3K, confirming its catalytic role (Fig. 2D).

Comparing ADP binding in HsFN3K with the 2.37 Å resolution AtFN3K-ADP crystal structure (PDBid 6OID), we observed that the nucleotide sugar in HsFN3K is in canonical C3'-endo geometry, while in AtFN3K structure the C2'-endo sugar geometry alters the α and β phosphate positioning in the catalytic site compared to HsFN3K. The altered nucleotide positioning allows the critical Lys41 to interact with the α-phosphate via a salt-bridge interaction (~ 2.8 Å apart) in HsFN3K, while it stays 3.9 Å apart in the AtFN3K structure. The β-phosphate placement in the two structures is also slightly different, however the nucleotide base-specific interactions are conserved. The positioning of Lys41 and the presence of a $Mg^{2+}$ ion in the active site in HsFN3K support the placement of the nucleotide in a catalytically competent geometry, with both α and β phosphates directly coordinating the $Mg^{2+}$ ion in an octahedral geometry along with Asp234 and Asn222 and two water molecules (Fig. 2C, Supplementary Fig. 4A). The α and β phosphate recognition by a conserved Lys or Arg residue is known to be a crucial feature of nucleotide recognition in members from different kinase families[38,39], and our structures confirm that FN3K also has a similar nucleotide recognition mechanism. Nevertheless, we cannot rule out the fact that the nucleotide positioning in the AtFN3K structure might be modeled imprecisely due to its lower resolution.

DMF sits in a negatively charged pocket in the C-lobe, close to the ADP β-phosphate. The DMF sugar moiety binds in its linear tautomeric form in the substrate binding site (Fig. 2B, C, Supplementary Fig. 4A), which is perfectly adapted to stabilize this active linear sugar geometry. The DMF binding site in HsFN3K is lined with several bulky

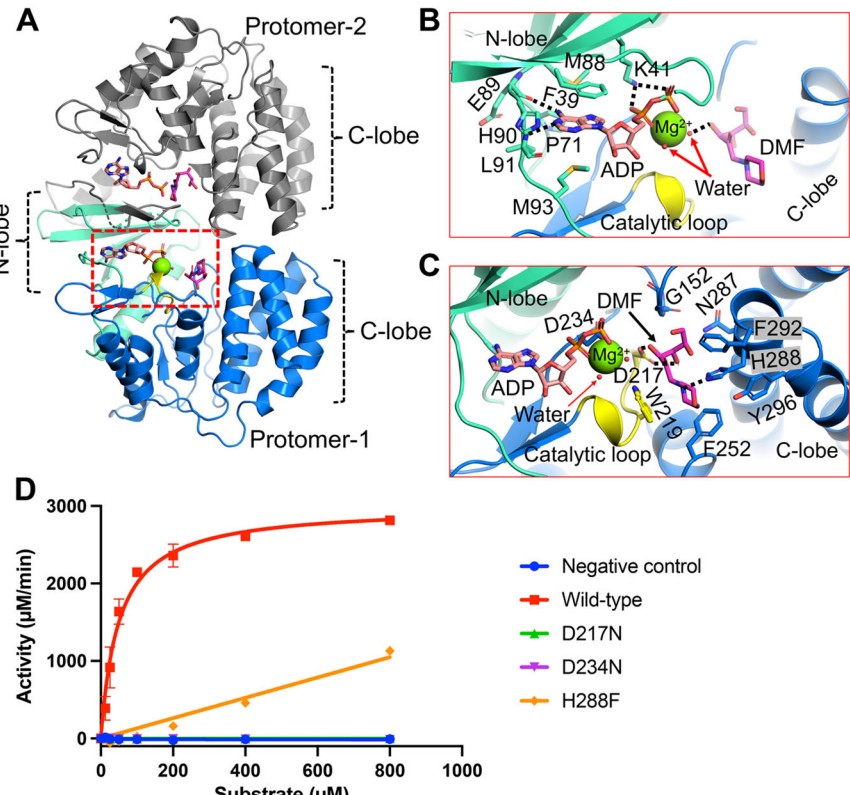

**Fig. 2 | ADP and DMF binding to HsFN3K. A** The crystal structure of HsFN3K in complex with ADP and the substrate DMF. Both protomers in the asymmetric unit contain ADP-DMF, while only protomer-1 shows a coordinated $Mg^{2+}$ ion (green sphere). The molecular interactions of the (**B**) ADP and the substrate (**C**) DMF observed in the crystal structure. The ADP base shows specific interactions and is surrounded by bulky hydrophobic amino acids. The conserved Lys41 binds both phosphates. The substrate binding pocket is lined with several aromatic residues including His288 (blue sticks) and Trp219 from the catalytic loop (yellow). The fructose sugar moiety interacts with the catalytic Asp217 (yellow sticks). The coordinating water molecules are shown as red nd-spheres. Direct interactions are shown as black dotted lines (**D**) In vitro kinase assays showing the phosphorylation of DMF by WT and several variants of HsFN3K. D217N and D234N show no kinase activity and the H288F variant shows significantly reduced kinase activity on DMF compared to the WT-HsFN3K. Data are presented as mean ± SD ($n = 3$). Source data are provided as a Source Data file.

aromatic residues, including Trp219, Phe252, Tyr296, Phe292, and His288, which interact with the aromatic morpholino group of DMF. His288 interacts directly with the morpholino group nitrogen via a H-bond (Fig. 2C, Supplementary Fig. 4D). Notably, this nitrogen is analogous to the primary amine nitrogen of the lysine side chain, which accepts the fructose moiety to form the fructose-lysine upon glycation. The recognition of this nitrogen by HsFN3K His288 is critical for substrate phosphorylation since its mutation to phenylalanine results in significantly reduced phosphorylation of DMF in vitro (Fig. 2D). The six-carbon sugar moiety adopts a curved geometry with four terminal carbons laying almost planer onto a tight turn containing Cys151-Gly152 in the substrate binding site. The sugar hydroxyl groups are exposed and interact with polar amino acids from the HsFN3K C-lobe (Fig. 2C, Supplementary Fig. 4D). Asp217 establishes H-bonds with the sugar moiety with both O3' and O4' atoms. This interaction is critical for HsFN3K function since a mutation of Asp217 to asparagine (D217N) completely abolishes HsFN3K kinase activity in vitro (Fig. 2D). O4' also interacts with Asn284 and Asn287 via a water molecule. Upon ATP hydrolysis, the phosphate molecule is transferred onto the O3' atom of the sugar moiety, and the crystal structure shows that the O3' atom is positioned closest to the ADP in the catalytic center, directly interacting with one of the water molecules in the $Mg^{2+}$ coordination sphere (Fig. 2B, Supplementary Fig. 4).

Another 1.8 Å resolution HsFN3KΔ structure in complex with ADP and DMF (labeled as (II) in Table-1) obtained in different soaking conditions exhibits almost identical interactions between ADP, DMF, and the enzyme. Overall, these structures of HsFN3K bound to the DMF substrate provide molecular insights into fructoselysine recognition.

## The pre-catalytic state in HsFN3K-mediated phosphorylation of a glycated substrate

To gain further insights into the phosphate transfer from ATP onto a glycated substrate, we determined the crystal structure of HsFN3KΔ bound with ATP and DMF (Table 1), representing a true pre-catalytic state. The overall domain-swapped PKL fold is conserved in the dimeric structure, with a partly disordered P-loop. Notably, both ATP and DMF are clearly observed in one FN3K molecule in the asymmetric unit, while only an ADP molecule is observed in the other FN3K protomer (Fig. 3A and Supplementary Fig. 5A–D). A lack of a bound $Mg^{2+}$ ion in the catalytic site likely prevented turnover, allowing us to clearly visualize both ATP and DMF in this subunit. The ATP fits perfectly in the nucleotide-binding pocket and is stabilized via a network of conserved interactions with neighboring amino acids and water molecules, as observed in the previous structure with ADP. The γ-phosphate of ATP is intricately coordinated in a pre-catalytic state via the catalytic Asp234, Asn222, Trp219, and the main chain amide group of Cys24 along with several water molecules (Fig. 3B). Interestingly, this highly coordinated γ-phosphate establishes direct H-bonding interactions with the phosphate-receiving O3' atom and the O2' atom of the fructose sugar moiety (Fig. 3B and Supplementary Fig. 5B).

We observed that with ATP binding in the catalytic site, Trp219 flips to directly interact with the γ-phosphate, while it predominantly adopts an alternative rotamer conformation in the ADP-bound FN3K

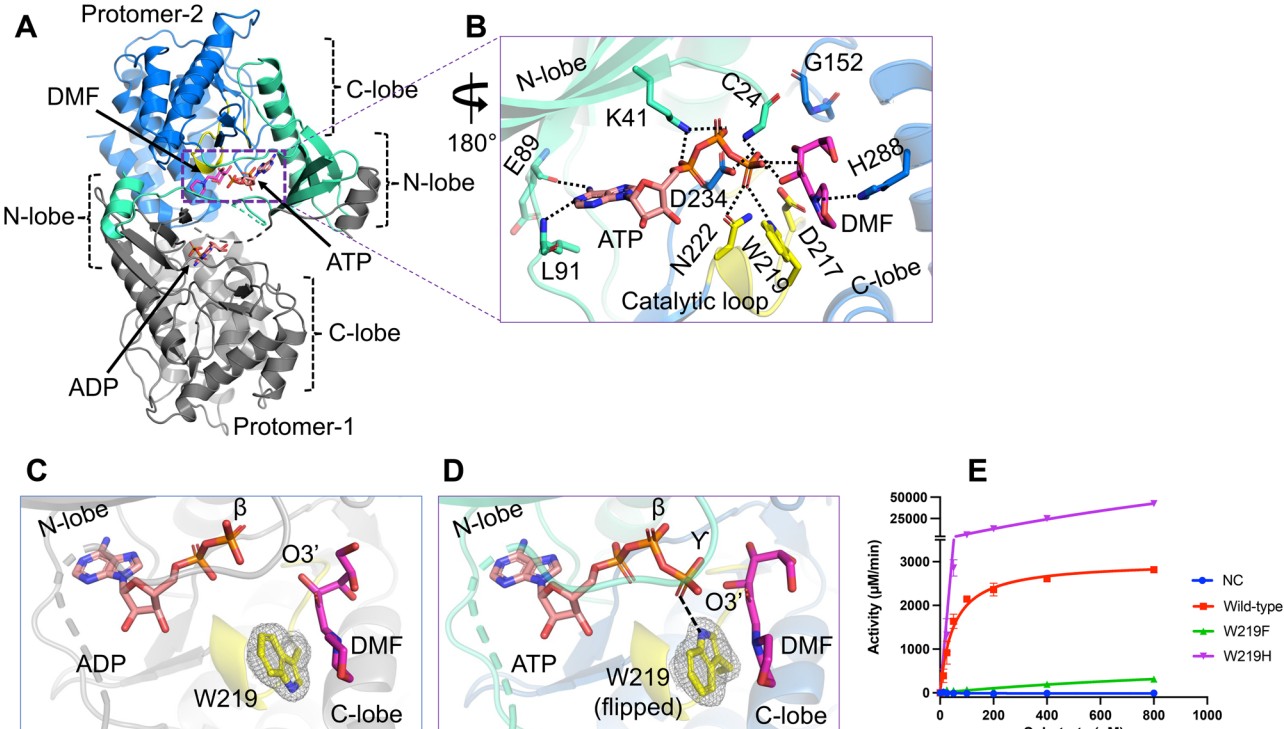

**Fig. 3 | The pre-catalytic state of HsFN3K mediated phosphorylation. A** The crystal structure of HsFN3K in complex with ATP and substrate DMF is shown in cartoon representation. Protomer-1 shows only ADP in the catalytic site, while ATP and DMF are both observed bound to protomer-2. **B** A close-up view of the molecular interaction of ATP and DMF bound to HsFN3K protomer-2. Direct interactions are shown as black dotted lines. The adenosine base and K41 interactions are conserved. Several interactions with the ATP γ-phosphate, including a direct interaction with the sugar moiety, are shown. W219 (yellow sticks) is flipped compared to the ADP-bond structure and establishes direct interactions with ATP, representing the true pre-catalytic state. The side-by-side comparison of ADP (**C**) and ATP (**D**) nucleotide binding in HsFN3K shows W219 in opposite orientations. The 2Fo-Fc electron density corresponding to W219 is shown as a mesh at 1.0 σ level with a carved radius of 1.6 Å. The protomer lacking the Mg²⁺ ion in the active site is shown for the ADP structure. The predominant alternative conformation of W219 is shown in ADP bound form. **E** In vitro kinase assays showing the phosphorylation of DMF by WT and W219 variant HsFN3K. W219H shows substantially elevated kinase activity on DMF compared to the WT, acting as a FN3K super kinase. Data are presented as mean ± SD (*n* = 3). Source data are provided as a Source Data file.

structure (in protomer-2) (Fig. 3C, D), suggesting its potential role in sensing the γ-phosphate before ATP hydrolysis. On the other hand, protomer-1 in the ADP-bound HsFN3K structure contains a Mg²⁺ ion, which partially occupies the γ-phosphate position in the active site, favors Trp219 to flip towards it to interact with a water molecule from the Mg²⁺ coordination sphere (Fig. 2C and Supplementary Fig. 5E) and is in a similar orientation as in the pre-catalytic state. This observation indicates that Trp219 flipping and γ-phosphate coordination are interlinked in HsFN3K.

Trp219 is highly conserved in the FN3K kinase family (Supplementary Fig. 5F) and a mutation of HsFN3K Trp219 to phenylalanine almost completely abolished the kinase activity on DMF (Fig. 3E), suggesting that the γ-phosphate recognition by Trp219 may represent a nucleotide triphosphate-sensing mechanism, important for its kinase activity. Notably, this residue is part of the kinase catalytic loop and is specific to the Fructosamine kinase family only[38,40], indicating that the molecular mechanism of ATP sensing in FN3K is different from other kinases. Therefore, targeting Trp219 in FN3K might provide a specific handle to regulate FN3K-mediated phosphorylation in cancer cells and other pathological conditions. Several members in other kinase families, including PI3K, maltose kinase (MalK), and MethylthioRibose kinase (MTRK), have a histidine residue conserved at this position in the catalytic loop[38,40]. We, therefore, also tested the effect of having a histidine residue at this position in FN3K. Interestingly, a W219H mutation converts HsFN3K into a super kinase, exhibiting several-fold higher kinase activity against DMF in vitro (Fig. 3E), with a significantly higher ATP turnover. To this end, we interrogated available TCGA data

for potential hotspot mutations at this residue but did not observe any. The lack of positive selection for a hyperactive FN3K variant might suggest deleterious effects that outweigh survival benefits in cancer cells.

## FN3K substrate binding site is dynamic
Next, we analyzed substrate binding in different HsFN3K structures bound with different nucleotides and found that the fructose sugar moiety undergoes significant structural changes upon ATP binding in the catalytic site compared to the ADP-bound structure (Fig. 4). Upon ATP binding, the γ-phosphate interactions with the sugar O2' and O3' atoms pull the sugar moiety closer to the catalytic site. This slight adjustment allows the O4', O5', and O6' atoms to reorient and adopt a different geometry altogether. The O4' atom, which points away from the nucleotide-binding site in the ADP-bound structure, flips towards the ATP γ-phosphate and establishes interactions with two extra water molecules in addition to its interaction with the conserved Asp217 (from the catalytic loop) and Gly152 main chain amine (Fig. 4B). The O5' atom, which exhibited no interactions in the ADP-bound structure, now interacts with Asn287 and a water molecule. O6' points toward the nucleotide-binding site in the ADP-bound structure, while it points away from the nucleotide-binding site in the ATP-bound structure, though it doesn't bind to any amino acids nor to any water molecules in either structure. Nevertheless, the interaction between His288 and the DMF morpholino nitrogen is conserved in the different HsFN3KΔ structures. Overall, the observed structural changes of the fructose sugar moiety in the substrate binding site of HsFN3K provide

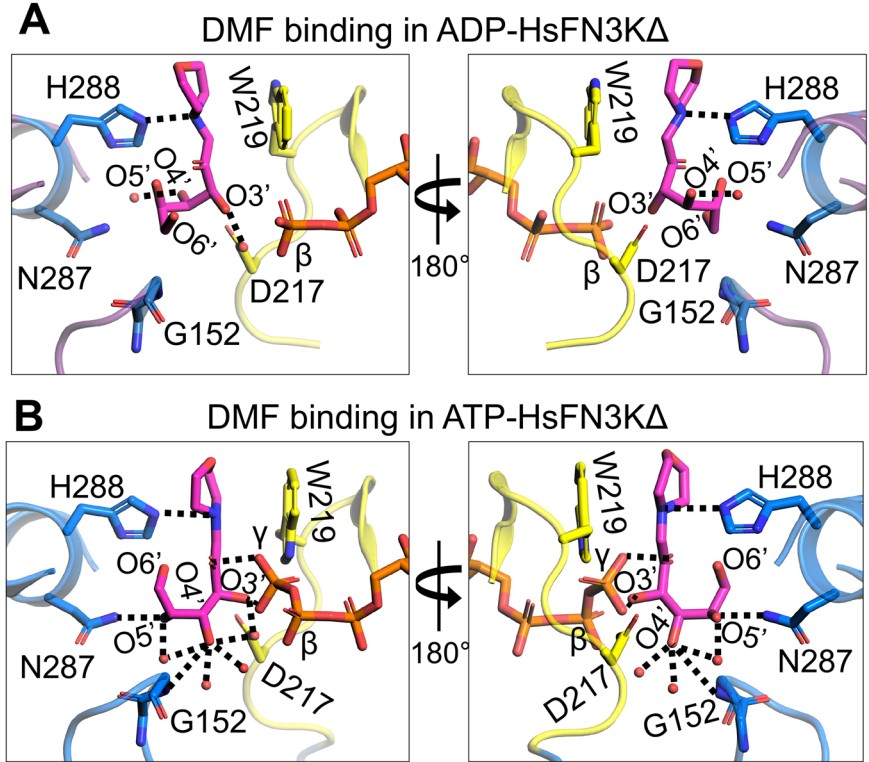

**Fig. 4 | Structural reorientation of the sugar moiety in the HsFN3K kinase cycle.** Close-up views of the fructose sugar moiety coordination in the (**A**) ADP and (**B**) ATP-bound structures. The sugar O3′ interactions are conserved in the two FN3K states. In the ATP bound pre-catalytic state, the geometry of the sugar moiety is stabilized by several water molecules and an interaction between O4′ and D217. The γ-phosphate interactions with sugar O2′ and O3′ atoms are also shown. Direct interactions are shown as black dotted lines.

mechanistic insight into substrate dynamics prior to phosphorylation by HsFN3K.

We also determined a 1.90 Å resolution crystal structure of HsFN3KΔ bound to the non-hydrolyzable ATP analog, AMPPNP with DMF (Table 1) (Supplementary Fig. 6A–C) and included it in our comparative analysis with the ADP and ATP bound HsFN3K structures described above. AMPPNP binds in the nucleotide-binding site in a similar fashion to ATP. However, its γ-phosphate is positioned slightly differently and does not facilitate Trp219 flipping (Fig. 5A and Supplementary Fig. 6D), which stays in a predominantly alternative conformation (~63%). Moreover, the conformation of the fructose moiety in the AMPPNP-bound HsFN3K structure is very similar to the ADP-bound state (Fig. 5B), rather than the ATP-bound pre-catalytic state of HsFN3K (Fig. 5C). Overall, this comparative analysis highlights that even upon AMPPNP binding, a lack of direct recognition of the γ-phosphate by Trp219, prevents the DMF sugar moiety from adopting a phosphate receptive geometry as observed in the ATP-bound HsFN3K structure (the true pre-steady state).

### FN3K-mediated phosphorylation of a glycated protein

To further investigate the activity of HsFN3K, we tested its in vitro kinase activity on glycated lysozyme protein substrates. To that end, we prepared glycated lysozyme by incubating 3.5 mM lysozyme with either 1 M glucose or 1 M ribose for 21 days and 7 days, respectively, followed by purification by size-exclusion chromatography (Supplementary Fig. 7A, B). The peak fractions collected from each preparation (Fig. 6A) were concentrated to ~10–13 mg/mL and stored at −80 °C. Glycated lysozyme showed very similar thermal melting spectra (Tm = 69.5 °C) as the WT lysozyme (Tm = 70.5 °C), suggesting that the lysozyme structure was not disturbed over the period of sugar treatment (Supplementary Fig. 7C). We verified the extent of glycation

using intact protein mass spectrometry analysis. As shown in Supplementary Fig. 8, this analysis confirmed the molecular weights of both the unmodified and the glucose-glycated lysozyme (Supplementary Fig. 8A–C). While the unmodified lysozyme produced a single ion (albeit in multiple charge states), the spectrum of the glucose-modified lysozyme clearly indicated the presence of multiple species within each charge group, separated by a mass difference (D-mass) of 162 Da. This Δ-mass is consistent with the addition of one glucose molecule (180 Da) to the protein (Supplementary Fig. 8B, C) via a condensation reaction with the loss of a water molecule (18 Da). The two groups, glucose-1 (Supplementary Fig. 8B) and glucose-2 (Supplementary Fig. 8C), represent similar glycated-lysozyme species, differing only in the number of unmodified lysozyme molecules. We, therefore, used the glucose-1 sample primarily in subsequent biochemical experiments. Unfortunately, the ribose-glycated lysozyme showed very heterogeneous populations in the various charge groups in the spectrum (Supplementary Fig. 8D).

To demonstrate the specificity of FN3K kinase activity against the glucose moiety on the protein substrate, we incubated purified HsFN3K with either unmodified lysozyme or the lysozyme glycated with glucose or ribose in the presence of γ-$^{32}$P ATP (Fig. 6B). For unmodified lysozyme, no γ-$^{32}$P signal was observed, indicating a lack of phosphorylation. However, the glucose-glycated protein was phosphorylated and showed a clear γ-$^{32}$P signal transferred by HsFN3K. Additionally, we observed relatively higher phosphorylation on the glucose-1 species compared to the glucose-2 species. Interestingly, although the ribose-glycated lysozyme suffered from high heterogeneity (Supplementary Fig. 8D), it was phosphorylated to a higher extent than the glucose-glycated counterparts. This result is consistent with the higher reactivity of ribose towards its glycation targets[33], which could result in more glycation on lysozyme in our study.

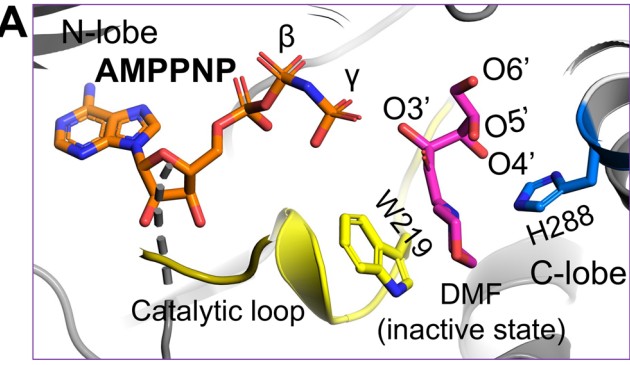

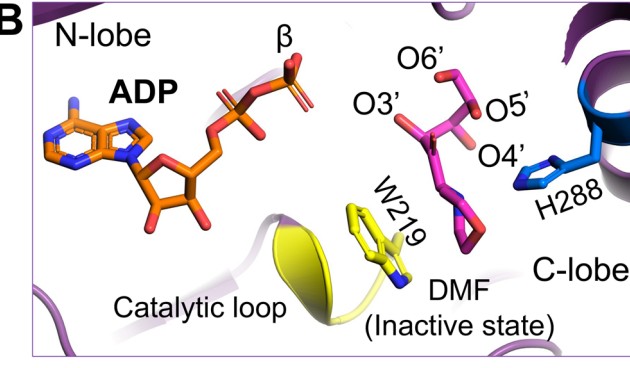

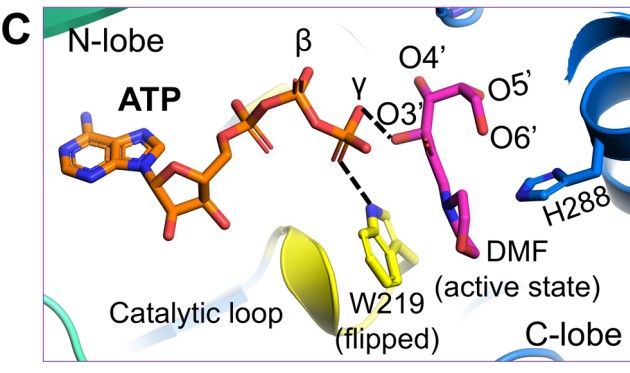

**Fig. 5 | The AMPPNP-bound pre-catalytic state of HsFN3K.** Close-up view of DMF substrate binding to HsFN3K in (**A**) AMPPNP, (**B**) ADP, and (**C**) ATP-bound structures. The β-phosphate in ADP is positioned far from the sugar moiety of DMF (inactive conformation). The AMPPNP γ-phosphate is positioned close to the DMF sugar moiety, but with W219 (yellow sticks) in a predominant unflipped alternative conformation, is unable to induce the sugar conformational changes. The flipped W219 interaction with the ATP γ-phosphate allows it to stably interact with DMF (magenta sticks) sugar O3' and triggers the active conformation. The H288 (blue sticks) interactions with the DMF morpholino group are conserved in all structures.

We verified the role of different residues identified from our structural analysis, crucial in phosphorylation by HsFN3K for these substrates as well. We tested HsFN3K variants of the nucleotide-coordinating Asp234, the sugar moiety coordinating Asp217, and the ATP γ-phosphate sensing Trp219 in an in vitro kinase assay on glycated lysozyme. As expected, all these HsFN3K variants were incapable of phosphorylating the glycated lysozyme, as opposed to wild-type HsFN3K (Fig. 6C). Interestingly, the hyperactive HsFN3K variant, W219H, only phosphorylated the glycated protein substrate, retaining its substrate specificity (Fig. 6D), suggesting that this hyperactivity is not equivalent to a loss of substrate specificity, or enzyme promiscuity, but instead represents a more active state of the kinase. Overall, our in vitro kinase assays with the glycated protein substrate align perfectly with the HsFN3K structural observations and biochemical assays

with the small molecule sugar mimic substrate, DMF, described above, further validating the structural observations.

To gain molecular insights into the post-catalytic state of FN3K, we sought to capture HsFN3K with a phosphorylated substrate in its binding site. We reasoned that the phosphorylated substrate would be unfavored and clash with Asp217 in the substrate binding site. We mutated Asp217 to a serine residue to create extra space for the phosphate while maintaining the overall negative charge in the binding pocket. We obtained crystals of the HsFN3KΔ_D217S variant in its apo form and performed soaking experiments using a mixture of an enzymatically prepared phosphorylated-DMF and ADP (see Methods). The overall kinase fold in this structure is almost identical to our previously described HsFN3KΔ structures (Fig. 7A), but the N-lobe α1-helices are not swapped in this structure, likely due to the different crystallization condition for HsFN3KΔ_D217S. This crystal structure does not show reasonable electron density for phosphorylated-DMF in the substrate binding site. Interestingly, in the nucleotide-binding site, a clear density for ATP (unused ATP from the mix containing phos-phorylated-DMF) was observed in both protomers in the asymmetric unit with a flipped Trp219 binding its γ-phosphate (Fig. 7B, C). The empty substrate binding site indicates that either the phosphorylated-DMF is too unstable to be trapped in the crystal or that the D217S mutation is unable to stabilize the phospho-DMF, which diffuses out of the binding site. Furthermore, a serine would be too far to H-bond with the O3' of the sugar moiety, explaining why DMF is absent in the substrate binding site in this structure. The presence of ATP in the nucleotide-binding site also suggests that ATP is preferred by HsFN3K over ADP. The melting temperature of HsFN3KΔ changes from 53.5 °C to 61.5 °C and 64 °C when incubated with ADP and ATP, respectively, in a thermal melting assay, confirming the preference for ATP over ADP (Fig. 7D). Moreover, we observed no added thermal stability to HsFN3K when DMF was incubated together with ADP or ATP (Fig. 7D).

Notably, in the HsFN3K-D217S structure, the distance between the nitrogen of the flipped W219 side chain and ATP γ-phosphate increases by -0.5 Å compared to the HsFN3K-ATP structure, due to the slight movement in the γ-phosphate positioning in the absence of the sugar moiety in the substrate binding site. This observation suggests that sugar binding also influences the nucleotide geometry in the binding site, and a true pre-steady state would not be attained in the absence of the substrate sugar in HsFN3K.

## Discussion

Kinases are excellent therapeutic targets due to their well-defined substrate binding sites and their involvement in a large number of critical cellular processes. The enzyme fructosamine-3-kinase (FN3K), which is known for deglycating hemoglobin[20], preventing micro- and macrovascular complications in type 2 diabetes mellitus (T2DM)[41], and macular degeneration by deglycating advanced glycation end pro-ducts (AGEs)[42], also exhibits potential oppurtunity for a therapeutic intervention.

Recently, FN3K was identified as an important upstream reg-ulator of the transcription factor NRF2, providing a proliferative advantage in liver and lung cancer cells[29]. In this regard, FN3K plays a role in the deglycation of NRF2, by phosphorylating the sugar moiety on glycated NRF2 residues, which leads to the sugar removal. Due to the elevated need for sugar for energy production, cancer cells are known to have upregulated sugar uptakes[43]. The increased sugar level causes non-specific glycation of proteins like NRF2. Glycated NRF2 has impaired transactivation activity, and deglycation restores its activity leading to NRF2 reactivation[29]. Although transcription factors like NRF2 have been challenging targets for therapeutic development, targeting associated kinases may offer an attractive solution. In this study, we have carried out a systematic structural-functional characterization of FN3K that can inform rational design of antagonists against the kinase.

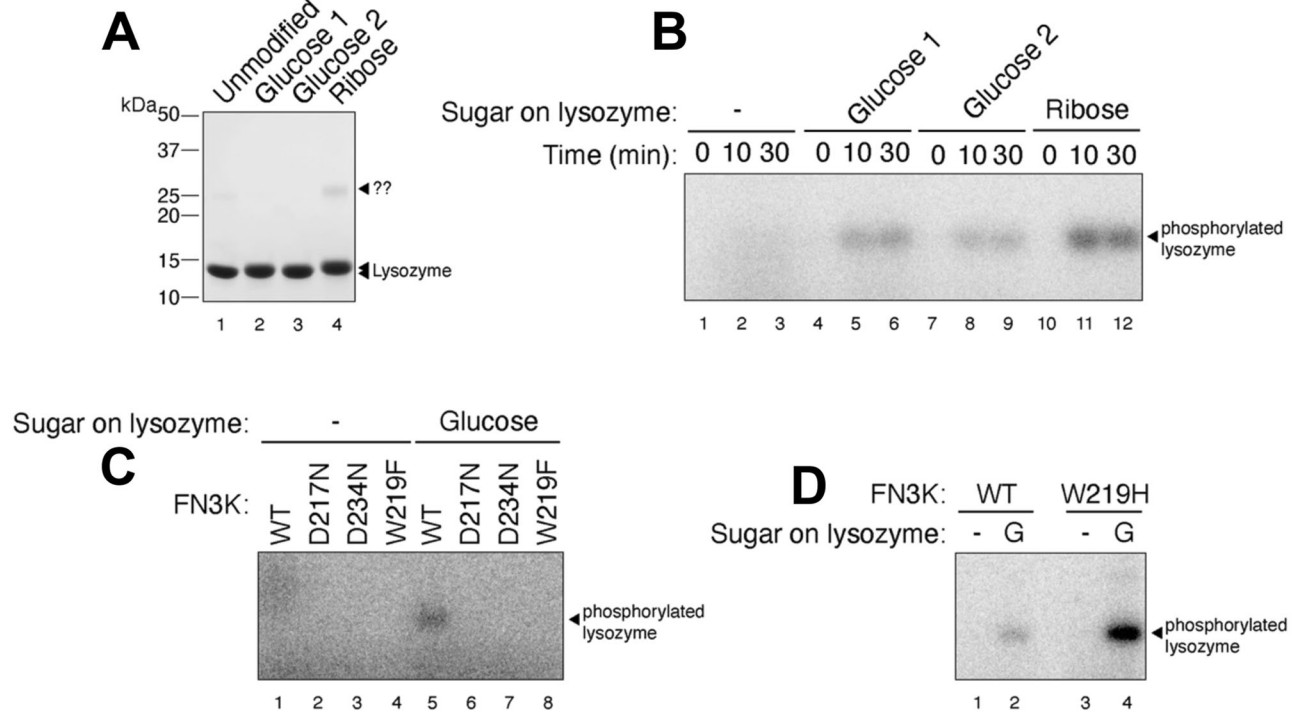

**Fig. 6 | In vitro phosphorylation of a glycated protein substrate. A** SDS-PAGE showing the purified glycated lysozyme substrate following in vitro glycation with glucose or ribose sugar. The protein ladder is also marked. The '??' represents an unknown contaminant in the lysozyme sample. **B** HsFN3K phosphorylates the glycated lysozyme only. No phosphorylation signal is observed for unmodified lysozyme within the 30 min time course. **C** The phosphorylation of glucose glycated lysozyme with different HsFN3K WT, D217N, D234N, and W219F variants. No variant HsFN3K showed a phosphorylation signal as compared to WT HsFN3K. **D** The HsFN3K W219H mutation exhibits substantially elevated kinase activity on the glycated lysozyme, without affecting specificity. A representative gel image is shown in different panel (*n* = 2).

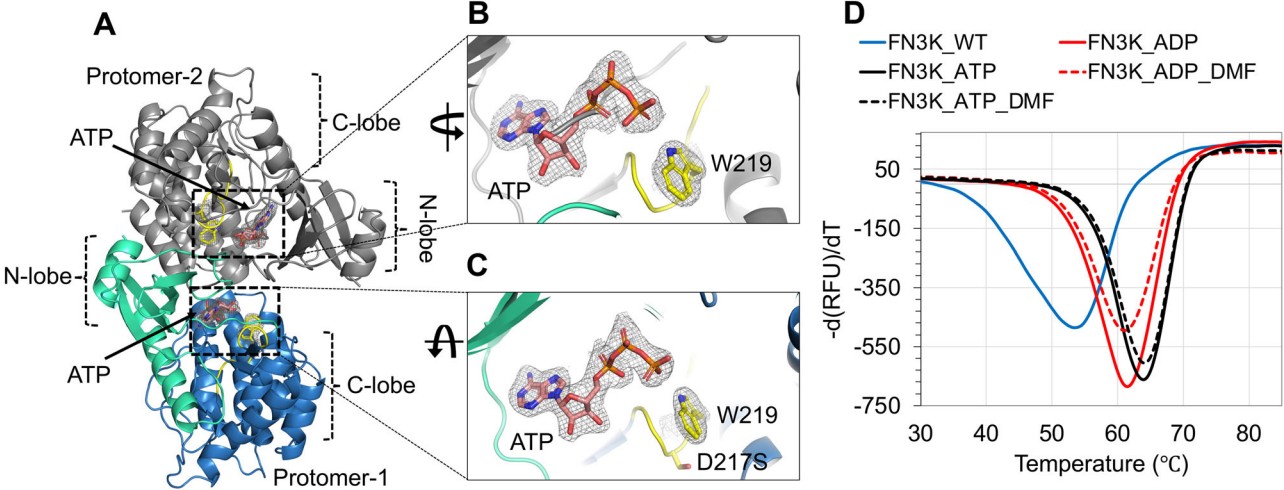

**Fig. 7 | Structure of HsFN3K-D217S. A** The crystal structure of the HsFN3KΔ_D217S variant showing the bound ATP nucleotide in both protomers in the asymmetric unit. The electron density of both ATP molecules and the flipped W219 is shown for (**B**) protomer-1 and (**C**) protomer-2 in the ASU. No substrate was observed in the substrate binding site. (**D**) The thermal melting spectra for HsFN3K-FL (WT) (blue solid lines) in the presence of DMF and different nucleotides. The analysis shows that the *Tm* change is significantly higher with ATP (red solid lines) compared to ADP (black solid lines). DMF binding shows no further *Tm* increase. Source data are provided as a Source Data file.

We show that HsFN3K is capable of specifically phosphorylating the fructosamine on the small molecule substrate DMF, as well as the sugar moiety on a protein substrate. Apo-HsFN3K has an overall PKL kinase fold, and although it was purified under reducing conditions as a primarily monomeric protein in solution (Supplementary Fig. 2A), it forms a domain-swapped dimer supported by an intermolecular disulfide bond in the P-loop in the crystal structure. In solution the dimeric form of HsFN3K is dependent on the redox state mediated via the conserved Cys24 residue (Supplementary Fig. 2G), similar to AtFN3K. Interestingly, the dimeric species was not observed when FN3K was purified from *E. coli*, implying that FN3K dimerization might be partly facilitated by either post-translational modifications or mediated by eukaryotic chaperones. In contrast to the previously observed domain-swapped AtFN3K dimer, which was reported to be

catalytically inactive[16], HsFN3K dimer is significantly more active than its monomeric species and has a distinct placement of the P-loop and the α1 helix of the N-lobe (Supplementary Fig. 3). In kinases, the P-loop is crucial for nucleotide sensing, and its involvement in FN3K dimerization having altered kinase activity might be a form of regulation in FN3K family members. Furthermore, the same study showed that HsFN3K activity increases under reducing conditions, suggesting that the HsFN3K dimer is less active than the monomer. This discrepancy with our work is likely due to the fact that we used reducing conditions during protein purification to better mimic physiological conditions. This allowed us to observe the redox state of HsFN3K with confidence, as the purified dimer gets completely reduced to monomer under reducing conditions (Supplementary Fig. 2F-G). Whereas the HsFN3K dimer in the previous study does not monomerize, suggesting a possible issue with the protein fold thus affecting its activity. Notably, the existence of a dimeric form of HsFN3K in vivo has yet to be confirmed.

Here, we provide a detailed molecular illustration of fructose-lysine recognition by HsFN3K. We revealed that several bulky aromatic residues line the substrate binding site, positioned around the DMF morpholino group, while the sugar moiety adopts a strict geometry stabilized by different polar residues and water molecules in the binding site. In the context of a glycated protein substrate, these aromatic residues would line the long aliphatic portion of a lysine or arginine side chain, while the fructose sugar moiety would access the pocket at the active site. Among the residues interacting with the fructosamine sugar moiety, Asp217 is absolutely critical for kinase function, whereas the direct interaction of the nitrogen atom from the basic amine of His288 is crucial for efficient substrate phosphorylation. His288-mediated fructosamine sensing might function as a checkpoint for HsFN3K-mediated substrate phosphorylation given the significantly reduced kinase activity on DMF and glycated substrates compared to WT-FN3K.

The different steps of the phosphorylation reaction by FN3K were visualized via the determination of structures of HsFN3K bound with different nucleotides. Several specific residues, including the catalytic Asp234 and the conserved Lys41, stabilize the ADP in the active site of HsFN3K, which is different from the position of the ADP observed in the AtFN3K structure (see Supplementary Fig. 3C). Upon binding of the ATP in the active site, an intricate network of H-bonds is established with the γ-phosphate (Fig. 3A, B), leading to improved thermal stability (Fig. 7D). The catalytic site in FN3K is negatively charged. However, the divalent cation probably facilitates the binding of the negatively charged nucleotide to this site. Similar electrostatic features exist in other kinases such as CLK3[44] and DYRK1a[45].

Notably, the FN3K kinase-specific Trp219 residue from the catalytic loop flips and binds the γ-phosphate, likely a critical checkpoint prior to hydrolysis. AMPPNP is regularly used with kinases to trap them in a pre-catalytic state geometry. Interestingly, though AMPPNP fits into the HsFN3K active site in a similar manner to ATP, it does not induce a true pre-catalytic state containing a flipped Trp219, due to the subtly different positioning of its γ-phosphate. Other protein kinase families have different residues at this position, often either a histidine or a phenylalanine. Interestingly, an HsFN3K W219H mutation converts HsFN3K into a highly active super kinase, while the FN3K W219F variant is catalytically inactive (Figs. 3E and 6C). This Trp residue is unique to FN3K kinases and might have implications in controlling the kinase activity of the enzyme. Targeting this FN3K-specific residue therapeutically might avoid cross-reactivity with other kinases in the cell.

Furthermore, we revealed the dynamic nature of the substrate binding site in HsFN3K, by visualizing different conformations of the fructosamine sugar in the presence of different nucleotides. Only in the ATP-bound pre-catalytic state with a flipped Trp219, does the sugar adopt a phosphate-receptive geometry with its O3' atom directly interacting with the γ-phosphate. In this conformation, the transferred γ-phosphate would directly bind to the sugar O3' atom by hydrolysis.

Overall our structural-functional analysis of HsFN3K paves the way for better understanding this unique family of kinases and serves as a foundation for the design of small molecule inhibitors against FN3K.

While our manuscript was under review, a study was published that describes the crystal structure of the human FN3K in the apo-state only[46]. From docking analysis, the authors also show that Asp217 and Asp234 play a catalytic role in FN3K kinase activity. Our study is consistent with these findings, but additionally provides crucial mechanistic insights into the binding and phosphorylation of the fructosamine substrate by FN3K.

## Methods

### Cloning, expression and purification of FN3K proteins

Full-length (FL) human (Hs) FN3K (Gene ID: 64122) and FN3K with an internal loop truncation (HsFN3KΔ: amino acid 117-138 replaced with a GSS linker) were sub-cloned into pET28a vectors to express an N-terminal 6xHis-TEV site fusion protein in *E. coli*. For insect cell expression, FL wild-type (WT) HsFN3K were sub-cloned into pFL vectors to express FN3K with a TEV cleavable N-terminal Twin-Strep-SUMO tag. Similarly, different HsFN3K variants (D217N, D234N, D217N/D234N, W219F, W219H, H288F) were sub-cloned into pFL vectors. Sub-cloning was performed using the sequence and ligase-independent cloning (SLIC) method. Primers used in FN3K cloning are listed in Supplementary Table-1 in the Supplementary Information file.

Bacterial expressions of FN3K were carried out by transforming Rosetta 2 (DE3) cells (Novagen) with the FN3K-pET28a plasmids mentioned above. Cells were grown at 37 °C until $OD_{600}$ reached 1.5, and were induced with 0.5 mM IPTG at 18 °C for 16 h before harvest. Cells were then resuspended in buffer-1 (50 mM HEPES pH 7.5, 200 mM NaCl, 10 mM imidazole, 2 mM ATP, 2 mM β-Me, 10% glycerol, protease inhibitors) and stored at -80 °C until purification. For insect cell expression of various FN3K constructs, we used a baculovirus expression system in Sf9 cells grown in HyClone CCM3 cell culture media (Cytiva). Sf9 cells were infected at 27 °C for 60 hr before harvesting in resuspension buffer-2 (25 mM Tris pH 8.0, 500 mM NaCl, 2.5 mM DTT, protease inhibitors) and stored at −80 °C until purification.

For bacterially-expressed FN3K purifications, the resuspended pellet was lysed by sonication and ultracentrifuged at 150,000 x g for 1 hr to separate the cleared lysate. Lysate was subjected to Ni-NTA affinity chromatography using the Ni-NTA agarose resin (Qiagen) equilibrated in Ni-EQ-buffer (50 mM HEPES pH 7.5, 200 mM NaCl, 10 mM imidazole, 2 mM ATP, 2 mM β-Me, 10% glycerol). After extensive washing with Ni-W-buffer (50 mM HEPES pH 7.5, 200 mM NaCl, 25 mM imidazole, 2 mM ATP, 2 mM β-Me, 10% glycerol), 6xHis-FN3K fusion protein was eluted in Ni-Elu-buffer (50 mM HEPES pH 7.5, 200 mM NaCl, 250 mM imidazole, 2 mM ATP, 2 mM β-Me, 10% glycerol). The Ni-NTA elutions were passed through a HiTrap SP-HP cation-exchange column (Cytiva) in IEX-buffer-1 (50 mM Tris pH 8.0, 0-1 M NaCl, 2 mM β-Me). Eluted FN3K was then incubated overnight with TEV protease (in 15:1 w/w ratio) to cleave the 6xHis-tag off. The next day, TEV-treated FN3K protein mix was subjected to size-exclusion chromatography on a HiLoad 16/60 Superdex 75 column (Cytiva) pre-equilibrated in GF-buffer (10 mM HEPES pH 7.5, 200 mM NaCl, 0.5 mM TCEP). Peak fractions were pooled, concentrated to 8.5 mg/mL, and were either used immediately for crystallization experiments or stored at −80 °C for future use. Typically, 6 L E.coli cell expression yielded 0.5 to 5.0 mg purified FN3K protein.

For FN3K purification from Sf9 cells, the resuspended pellet was sonicated and ultracentrifuged at 150,000 x g for 1 hr to separate the cleared lysate. Clarified lysate was then subjected to affinity purification with Strep-Tactin Superflow high-capacity resin (IBA lifesciences) pre-equilibrated in Strep-EQ-buffer (25 mM Tris pH 8.0, 500 mM NaCl, 2.5 mM DTT). After extensive washing of the resin (25 mM Tris pH 8.0,

150 mM NaCl, 2.5 mM DTT), Twin-Strep-SUMO-tagged FN3K was eluted from the resin (25 mM Tris pH 8.0, 500 mM NaCl, 5 mM desthiobiotin, 2.5 mM DTT). The eluant was incubated overnight with TEV protease (in 15:1 w/w ratio) to cleave the Twin-Strep-SUMO tag off. On the following day, TEV-cleaved products were subjected to cation-exchange chromatography (5 ml HiTrap SP-HP column from Cytiva) in IEX-buffer-2 (50 mM Tris pH 8.0, 0-1 M NaCl, 2.5 mM $\beta$-Me). The eluted FN3K was pooled, concentrated, and subjected to size-exclusion chromatography on a HiLoad 16/60 Superdex 75 column (Cytiva) pre-equilibrated in GF-buffer (10 mM HEPES pH 7.5, 200 mM NaCl, 0.5 mM TCEP). Monomeric and dimeric peaks from the gel-filtration run were separately pooled and concentrated to 5 mg/mL and stored in GF buffer supplemented with 10% glycerol at −80 ºC for future use. Typically, 3 L of Sf9 cell expression volume yielded 2.5 to 5.0 mg purified FN3K protein.

## Substrate preparation for kinase assay
For in vitro kinase assays, we first prepared a glycated form of lysozyme. For that, 0.5 g chicken egg white lysozyme (Sigma) was dissolved in 25 mM HEPES pH 7.3 containing 1 M glucose. This mixture was filtered through a 0.22 μm syringe filter and then incubated at 37 ºC for 21 days. No precipitation was observed after 21 days. The reaction mixture was injected into a HiLoad 16/60 Superdex 75 size-exclusion chromatography column (Cytiva) pre-equilibrated in 10 mM HEPES pH 7.5, 50 mM NaCl. The peak fractions were recovered, concentrated to 10 mg/mL, aliquoted, and stored at −80 ºC for future use. Glycations on lysozyme were verified by intact MS analysis before using it in assays. Unmodified lysozyme was prepared by directly dissolving chicken egg white lysozyme (Sigma) in the GF buffer (10 mM HEPES pH 7.5, 50 mM NaCl), concentrated to 10 mg/mL, aliquoted and stored at −80 ºC. The glycated lysozyme aliquots were mixed with glycoprotein denaturation buffer (NEB) and were denatured at 100 ºC for 10 min. The resultant mixture was spun at 20,000 x g at 4 ºC for 5 min to remove any insoluble aggregation, before using it in the in vitro kinase assays. Additionally, we used a known small-molecule sugar mimic, 1-Deoxy-1-morpholino-D-fructose (DMF) (Sigma) as a FN3K substrate in kinase assays.

## FN3K kinase assays
For non-radioactive kinase assays, 1 μM WT or variant HsFN3K in kinase buffer-1 (12.5 mM HEPES pH 7.5, 150 mM K glutamate, 5 mM Mg(OAc)$_2$, 0.01% NP-40, 1 mM DTT, 0.1 mg/mL BSA, 2.5% glycerol) was mixed with 1.6-2.6 units/mL pyruvate kinase (PK), 2.4-3.7 units/mL lactate dehydrogenase (LDH) (Sigma-Aldrich), 5.3 mM phosphoenolpyruvate (PEP) (Sigma-Aldrich), 0.2 mM NADH (Sigma-Aldrich) and 500 μM ATP in a 96-well, half-area, clear bottom microplate (Greiner Bio-One). A serial 2-fold dilution of DMF (800 μM – 2.5 μM) was prepared separately and rapidly mixed into the rest of the reaction components containing HsFN3K to initiate the kinase reaction. The continuous reduction in [NADH] levels in the reaction, resulting from its usage during ATP regeneration by the PK/LDH/PEP mix, was monitored at 340 nm wavelength with a Biotek Synergy Neo2 multi-mode reader (Agilent) at 25 ºC to measure the ATP utilization in the reaction for 45 min.

For kinase assays involving radiolabeled ATP, 4 μM denatured form of either unmodified or glycated lysozyme was mixed with 2 μM WT or variant FN3K in kinase buffer-2 (12.5 mM HEPES pH 7.5, 150 mM K glutamate, 5 mM Mg(OAc)$_2$, 0.01% NP-40, 1 mM DTT, 2.5% glycerol) containing ~50 μM ATP (radioactive γ-$^{32}$P-ATP and non-radioactive ATP). This reaction mixture was incubated at 37 ºC for 30 min (unless stated otherwise). Reactions were quenched by the addition of 0.5 mM EDTA and 4x SDS-PAGE sample buffer. Samples were then denatured at 100 ºC for 5 min, and analyzed on 15% NuPAGE gels. Gels were exposed to a phosphor screen overnight and autoradiographic imaging was performed using Typhoon FLA 7000 (Cytiva).

## Glycation and deglycation assay for NRF2 peptide
For the glycation assays with the NRF2 peptide (Eurogentec, 99.04% purity by HPLC), 50 μM peptide was incubated with excess D-ribose (100 mM) in 1X PBS at 37 °C. The reaction was quenched with an equivalent volume of HPLC buffer (45% acetonitrile in 0.1% aq formic acid) before being subjected to UPLC-MS analysis at daily intervals for 12 days. For the deglycation assays with NRF2 peptide and FN3K, 50 μM of D-ribose-glycated peptide was incubated with 100 nM recombinant WT HsFN3K in the reaction buffer (10 mM HEPES, pH 7.5, 20 mM NaCl, 0.5 mM TCEP, 10 mM MgCl$_2$) in the presence or absence of 2 mM ATP at 37 °C. The reaction was quenched with an equivalent volume of HPLC buffer and subjected to UPLC-MS analysis at 3-hour intervals.

UPLC–MS was carried out on a Waters Acuity SQD LC-MS in electrospray ionization (ESI) mode, with a Waters Acuity UPLC BEH C18 reverse-phase column (10 cm × 2.1 mm, 1.7 μm, 130 Å), using a flow rate of 0.3 mL/min and a gradient of 5–60% acentonitrile in 0.1% aq trifluoroacetic acid over 8 min. Analysis of UPLC-MS data was carried out using Waters MassLynx software. Raw UPLC-MS data was exported and analyzed using GraphPad Prism.

## Mass spectrometry measurements of glycated lysozyme
Pure lysozyme or purified glycated lysozyme were analyzed by top-down mass spectrometry using a Thermo Fusion Lumos orbitrap mass spectrometer interfaced with a nano electrospray ion source with a Thermo EasyLC1200 chromatograph. Samples were loaded on a 30 cm×100 μm ID column packed with Reprosil 20 C8 silica particles (Dr. Maischt), and resolved on a 5-55% acetonitrile gradient in water (0.1% formate) at 500 nl/min. Eluates were ionized by electrospray (2200 V) and transferred into the mass spectrometer, set to continuously record 240,000 resolution scans (m/z 750-3500 Th, max injection time 50 ms, max number of charges 1,000,000). Spectra were averaged over the entire chromatographic peak, and m/z values were deconvoluted to mass using the Thermo FreeStyle software v.1.7. The molecular weight of the unmodified lysozyme was empirically determined from untreated samples, and glycated isoforms were annotated based on the expected mass shift produced by the adducts. No statistical analysis was performed ($n = 1$). M/z and mass spectra were exported to Adobe Illustrator to generate figures.

## Protein thermal melting assays
For thermal melting assays 40 μg of different FL-HsFN3K proteins were mixed with Protein Thermal Shift Dye (ThermoScientific) in a thermal melting buffer (25 mM HEPES pH 7.5, 200 mM NaCl, 2 mM DTT) on ice. For the HsFN3K thermal melting assays testing different nucleotides and DMF, 30 μg of WT-HsFN3K protein was used, and the melting buffer was supplemented with 5 mM CaCl$_2$ to compete off the Mg$^{2+}$ ion in the catalytic site[47]. The protein:nucleotide and protein:DMF molar ratios were 1:1.5 and 1:2.25, respectively. For thermal melting experiments analyzing the lysozyme and the glycated lysozyme, 40 μg of each protein was used. Protein melting was measured over a temperature range of 15 °C to 95 °C using the Bio-RAD CFX Opus 96 Real-Time qPCR system. $n = 3$ for all experiments.

## Limited proteolysis
Limited proteolysis experiments for WT FL-HsFN3K protein were done using either Thermolysin (Promega) or Chymotrypsin (Sigma) proteases. ~5 μg of purified HsFN3K protein was mixed with the protease in 1:100, 1:250, 1:500, 1:1000 w/w ratios in Lim-Pro buffer (20 mM Tris pH 8.0, 200 mM NaCl, 2 mM CaCl$_2$, 5% Glycerol, 1 mM MgCl$_2$, 1 mM DTT). Reactions were incubated for 30 mins at 37 °C, before analyzing on 4-20% mini-PROTEAN TGX gel (BioRad). The observed gel bands were cut out and analyzed by mass-spectrometry to identify the cleavage sites, which aided in designing the FN3KΔ (amino acid 117-138 replaced with a GSS linker) constructs for crystallization.

## Micro Kit SEC analysis for FN3K

Different HsFN3K proteins were analyzed by injecting 20 μg (~ 28 μM) of purified protein onto a Superdex 200 increase 3.2/300 column (Cytiva) pre-equilibrated in MicroKit SEC buffer (HEPES 25 mM pH 7.5, NaCl 150 mM), which was further supplemented with 5 mM DTT for SEC under reducing condition. The proteins were incubated in the SEC buffer for 20 min before injecting onto the column. Absorption at UV280 and UV260 was recorded and used to plot the chromatograms.

## Non-reducing SDS-PAGE analysis

5 μg of purified HsFN3K proteins were incubated with or without 1 mM DTT into MicroKit SEC buffer in 12 μl volume at room temperature. After 20 min, 4 μl of 5X SDS-PAGE sample buffer without reducing agent was mixed in the protein sample, analyzed on 4-20% mini-PROTEAN TGX stain-free gel (BioRad), and imaged using BioRad ChemiDoc imaging system.

## Preparation of phosphorylated-DMF for soaking experiments

4 μM WT FN3K in kinase buffer-3 (10 mM HEPES pH 7.5, 200 mM NaCl) were mixed with 1 mM 1-Deoxy-1-morpholino-D-fructose (DMF), 3 mM ATP and 0.25 mM TCEP. This reaction mix was incubated at 20 °C for 16 h. The phosphorylation reaction was then incubated with 40 units of Proteinase K (NEB) at 50 °C for 45 min. An equal volume of phenol/chloroform/isoamyl alcohol (25/24/1, pH 6.6, Life Technologies) was added to this mix, vortexed for 1 min and spun at 20,000 x g for 5 min. The aqueous phase was isolated, aliquoted, and stored at −80 °C.

## Protein crystallization

Since we were not successful in obtaining crystals of FL-HsFN3K, all crystallization experiments were conducted using either HsFN3KΔ or HsFN3KΔ (D217S) variant. Crystallizations were performed at 20 °C using the hanging drop vapor diffusion method by mixing equal volumes of HsFN3K protein solution and the reservoir buffer to a final drop size of 400 nL.

HsFN3KΔ in its apo form was crystallized by mixing protein at 8.5 mg/ml with 0.5 M ammonium sulfate, 0.1 M tri-sodium citrate, and 1 M lithium sulfate. The crystals were harvested and cryo-protected using 2.5 M ammonium sulfate + 20% glycerol and flash frozen in liquid $N_2$.

HsFN3KΔ (D217S) variant was crystallized by mixing the protein at 7 mg/ml with 30% PEG 6000, 0.1 M HEPES pH 7.5, and 0.175 M $LiSO_4$. The harvested crystals were soaked in 0.9X reservoir buffer + 20% ethylene glycol supplemented with 5 mM Phospho-DMF-ADP mixture (enzymatically synthesized in-house) and flash frozen in liquid $N_2$.

For the HsFN3K_AMPPNP-DMF structure, apo HsFN3KΔ crystals were soaked in cryo-protectant solution (0.9X reservoir solution + 20% ethylene glycol) containing 1 mM AMPPNP and 10 mM DMF for 25 mins before flash freezing in liquid $N_2$.

For the HsFN3K_ADP-DMF (I) structure, apo HsFN3KΔ crystals were soaked in a cryo-protectant solution (0.9X reservoir solution + 20% ethylene glycol) containing 10 mM DMF, 5 mM $MgSO_4$ and 2.5 mM ADP-AlF$_3$ for 8 mins before flash freezing in liquid $N_2$.

For the HsFN3K_ATP-DMF, apo HsFN3KΔ crystals were soaked in cryo-protectant solution (0.9X reservoir solution + 20% ethylene glycol) containing 5 mM phospho-DMF-ADP mixture (enzymatically synthesized in-house) for 8 mins before flash freezing in liquid $N_2$.

For the HsFN3K_ADP-DMF (II) structures, apo HsFN3KΔ crystals were soaked in cryo-protectant solution (0.9X reservoir solution + 20% glycerol) containing 5 mM phospho-DMF-ADP mixture (enzymatically synthesized in-house) for 1 h 45 mins before flash freezing in liquid $N_2$.

## Data collection, structure determination, and refinement

The X-ray diffraction data for apo, AMPPNP-DMF, ATP-DMF, ADP-DMF (II), and D217S-ATP crystals were collected at the Advanced Photon Source NE-CAT section beamline 24-ID-E (λ = 0.979 Å) at Argonne National Laboratory. All the datasets were then integrated and scaled using its on-site RAPD automated programs (https://rapd.nec.aps.anl.gov/rapd/). X-ray diffraction data for the ADP-DMF(I) structure were collected at beamline 17-ID-1 (λ = 0.920 Å) at NSLS-II at Brookhaven National Laboratory and processed using XDS[48]. All crystals belonged to space group $P2_12_12_1$ with two molecules in the asymmetric unit. The apo structure was determined by Molecular Replacement (MR) using the N-terminally truncated *Arabidopsis thaliana* (At) FN3K structure (PDBid 6OID) as a search model[16]. The N-terminal region of apo HsFN3K was manually built using Coot[49] and refined using Phenix[50]. The atom contacts and model geometry were validated using the MolProbity server[51] and structure figures were generated using PyMOL (Version 2.5.5, Schrodinger, LLC). The APBS tool[52] was used for calculating the electrostatic surface potential.

The apo HsFN3K structure was used as a search model for MR to determine the structures of the complexes. All structures were refined using a similar approach as mentioned above.

Data collection and refinement statistics for all the HsFN3K structures are summarized in Table-1.

## Reporting summary

Further information on research design is available in the Nature Portfolio Reporting Summary linked to this article.

## Data availability

The atomic coordinates and structure factors have been deposited in the Protein Data Bank under accession codes 9CX8 for HsFN3K-apo, 9CXV for HsFN3K-ADP-DMF (I), 9CXW for HsFN3K-ADP-DMF (II), 9CXM for the HsFN3K-ATP-DMF, 9CXN for HsFN3K-AMPPNP-DMF and 9CXO for HsFN3K-(D217S)-ATP structure. Source data are provided with this paper.

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

## Acknowledgements

We thank Hans-Guido Wendel and members of Joshua-Tor laboratory for valuable discussions, and the CSHL core-proteomics facility for support with mass-spectrometry analysis. We thank the support of the beamline

staff at National Synchrotron Light Source II (AMX-17-ID-1), a U.S. Department of Energy (DOE) Office of Science User Facility operated for the DOE Office of Science by Brookhaven National Laboratory under Contract No. DE-SC0012704. We highly appreciate the support of the beamline staff at Northeastern Collaborative Access Team (NE-CAT) beamline 24-ID-E, funded by NIH grant GM P30G124165 and operated for DOE Office of Science by Argonne National Laboratories under contract DE-AC0206CH11357, for help with X-ray data collection. This work was supported by STARR Grant #36210201 (to Y.D. and L.J.). The mass spectrometry Shared Resource was supported by the CSHL Cancer Center Support Grant #5P30CA045508. L.J. is an investigator of the Howards Hughes Medical Institute.

## Author contributions

A.G., K.F.O., E.E. and L.J. conceived the study. A.G., E.E., and K.F.O. did the construct designs. A.G. and K.F.O. performed the crystallization and X-ray data collection. K.F.O. purified the proteins and did the biochemical assays. A.G. determined the crystal structures and biophysical assays. A.G., K.F.O. prepared the glycated lysozyme samples and P.C. performed the mass-spectometry analysis. Y.X. performed the kinase assay with the peptide substrate in Y.D. supervision. A.G., K.F.O., Y.D. and L.J. analyzed the data and wrote the manuscript.

## Competing interests

The authors declare no competing interests.
