## [Transparent Peer Review file · Nature Communications]

The Molecular Basis of Human FN3K-Mediated Phosphorylation of Glycated Substrates

Corresponding Author: Professor Leemor Joshua-Tor

Version 0:

Reviewer comments:

Reviewer #1

(Remarks to the Author)

The paper by Garg et al presents the crystal structure of fructosamine-3-kinase. This enzyme has the unique function of causing the removal of fructosamines (and other ketoamines) from proteins. The study provides not only information on the general structure of the protein, but also on the location of ATP and ADP and the fructosamine substrate in the catalytic site, allowing to identify the residues that are important for catalysis. The importance of these residues has been assessed by site directed mutagenesis and the capacity of the enzyme to cause deglycation of a ketoamine substrate has been assessed. Of particular interest is the role of a tryptophan residue (W219), which seems to limit the catalytic capacity of this enzyme.

A study on the structure of the homologous enzyme of *Arabidopsis thaliana* has previously been reported. There are some significant differences between the two structures and on the role of a conserved intersubunit disulfide bond. Furthermore, the new study provides much more information on the location of the substrate and the catalytic mechanism.

The work is well performed and the study very well described and represents a major progress in the field of deglycation and protein repair. I have the following minor comments to make.

1. In terms of substrate specificity, fructosamine 3-kinase (FN3K) differs from (human) fructosamine-related protein (FN3KRP) by its capacity to phosphorylate fructosamine (which have the phosphorylated 3' hydroxyl group in 'L' orientation), while both FN3K and FN3KRP phosphorylate ketoamine substrates (ribulosamine, psicamine) with a 3' OH group in D orientation (Collard et al. 2003). Can this difference in the specificity be ascribed to specific residues in the catalytic site ?
2. The choice of the peptide sequence that is studied as a substrate in figure 1A is a bit odd, since the physiological arginine in NRF2 is replaced by a lysine, to facilitate glycation. Actually, I am not sure that there is any paper in the literature showing convincingly (through a structural approach) that a ketoamines can be formed when aldoses react with an arginine. Other glycation products are formed, nonetheless. I want to stress that this is a minor point and that I am not recommending the authors to retry a more physiological substrate. The interest of the experiment is to confirm that FN3K is able to catalyze a deglycation reaction and this is convincingly shown. It may be useful to mention that the use of ribose rather than glucose as a glycating agent facilitated the demonstration of deglycation, since ribulosamine-3-phosphate (half-life \approx 20 min) is much less stable than fructosamine-3-phosphate (half-life \approx 8h).
3. The limitation of the catalytic activity of FN3K by the presence of a tryptophan at position 219 rather than a histidine may be useful to slow down side activities of this enzyme.

Reviewer #2

(Remarks to the Author)

The manuscript from Garg et al. describes the structural characterization of the FN3K enzyme that plays important roles in the regulation of protein glycation through direct phosphorylation of the damage to promote its spontaneous removal. The authors obtain several different crystal structures of FN3K in complex with different small molecules including a model of the glycation damage. Overall the structures appear high quality and support the author's conclusions. I am supportive of publication by have two minor comments.

Comment #1 - The arrow pushing mechanism by where elimination of the phosphorylated hydroxyl-group to regenerate an aldehyde at the 1-position, followed by hydrolysis of the glycation damage is not immediately obvious to me. Do the authors have a proposed mechanism? I was also unable to readily find one by searching the internet.

Comment #2 - How do the authors know which glycation product they are detecting my MS is the schiff-base and which is the Amadori product? I understand that one has a different retention time, but how do they sure which is which?

Reviewer #3

(Remarks to the Author)

The manuscript by Garg et al. describes the a series of crystal structures of the enzyme Homo sapiens fructosamine-3 kinase (FN3K) in the unliganded state and in complex with nucleotide analogs and a sugar substrate mimic to identify structural determinants of substrate recognition and catalysis. The structural findings were validated for selected active-site residues by mutagenesis. The activity of the wild-type and variants were tested on a small-molecule substrate but was also tested on glycated lysozyme. The findings show that the enzyme is active on protein substrate and that the activity is also sugar specific. The results should be of interest to those in the areas of cell signaling, protein kinases, drug discovery, and enzyme mechanisms. The human enzyme differs from the other structurally characterized plant enzyme which increases the impact of the work. To this point, and as described below, the authors should better analyze their findings on the dimeric state of the enzyme with the literature describing a proposed regulatory role of dimerization in the Arabidopsis enzyme. There are several points where the analysis of the findings and presentation can be improved. The authors may wish to consider the following points (page numbers from pdf for review):

Page 4- The authors state "and determined a series of crystal structures of human (Hs) FN3K in the apo, and in substrate-bound forms with various nucleotides representing different catalytic-transition states." These substrates and analogs are all ground state, there are no transition-state analogs crystallized. This should be restated.

Page: 6- In protein preparation in methods- could some idea of protein yield be given for those trying to reproduce the work (for instance mg protein/g cell paste for bacterial cell expression).

Page: 12-The authors state "We also observed a dimeric FN3K species during the purification from insect cells (Figure S1A-B), which exhibited ~60% higher kinase activity on DMF as compared to the monomeric FN3K species (Figure 1F)." It is very unusual for a monomer-dimer equilibrium to be so slow that the dimeric and monomeric species can be isolated. Upon separation, they will re-equilibrate. There is no discussion of this point and this left me very confused. Later in the manuscript it is explained that there is a disulfide bond between subunits which means that the monomer-dimer equilibrium is dependent on redox state. This needs to be at least summarized here and can be expanded upon later when the structure is described. Otherwise it is too confusing to the reader.

Page: 14- The authors state – "The electrostatic surface of the Apo-HsFN3K structure exhibits a negatively charged pocket near the P-loop for ATP binding, and a sulfate ion is observed in one protomer occupying the space for the nucleotide β -phosphate in this structure." It seems odd for an ATP binding pocket to be negatively charged as ATP itself is negatively charged. Is this common in kinases with this fold? Can the authors explain this for the reader?

When I looked at the comparison with the *A. thaliana* structure and the paper describing it I note that there is a completely different interpretation of the presence of the disulfide bonded dimer- In that paper (ref 16 here) the authors state "we found that the P-loop is stabilized in an extended conformation by a Cys-mediated disulfide bond connecting two chains to form a covalently linked dimer in which the reduction of disulfides resulted in AtFN3K activation. Consistent with this, HsFN3K, in which the P-loop Cys is conserved, was redox-regulated and displayed altered oligomerization when proliferating cells are exposed to acute oxidative stress." The opposite is found here, where the monomeric form is less active. The authors should address this important difference during the discussion. It is only briefly mentioned in the discussion section. A more in-depth description of the differences in structure and the accessibility of the active site should probably be given. If space is needed I would suggest removing or shortening the long paragraph on the attempt to obtain a structure with phosphorylated-DMF, which was unsuccessful.

Page: 15-The authors describe the differing geometries of ADP in this and the *A. thaliana* structure but do not address the importance. This is an example of a tendency for the results and discussion to be too descriptive. The authors should look to describe the mechanistic impact of the differing structures.

Page: 18- The authors state "Interestingly, a W219H mutation converts HsFN3K into a super kinase, exhibiting several-fold higher kinase activity against DMF in vitro (Figure 3E), with a significantly higher ATP turnover." As the authors kinetics show substrate saturation and they know the enzyme concentration, it seems that an analysis to yield k_{cat} and K_m is feasible for WT and mutants and would present more quantitative results.

Page: 24- Obviously the enzyme can turn over in crystallo, because of the observation of ADP in the binding site of crystals prepared with ATP and substrate DMF in one subunit. Do the author have a hypothesis as to why the other subunit is poised for catalysis but does not turnover?

Page: 36- The names of the complexes in Methods are not consistent with those in Table I or the results section of the text. I

was looking for the conditions for the ATP-DMF complex and could not find them.

Typos, grammatical errors

Page: 2- "to impact the cellular function of target protein" should read "to impact the cellular function of the target protein"

Page 2 "of HsFN3K in apo-state," should read "of HsFN3K in the apo-state"

Page: 3- "Alternatively, the sugar moiety on glycosylated residues can be phosphorylated by a class of small-molecule kinases called fructosamine-3-kinase (FN3K), which are present in mammals, birds, and plants" The enzyme name should probably be plural to read "fructosamine-3-kinases, which are present ...". then define the abbreviation in the next paragraph - which is actually already done.

Page: 4- "its phosphorylation activity is believed to be crucial for its regulation on NRF2" should read ".. to be crucial for its regulation of NRF2"

Page: 10- The authors state "all crystallization experiments were conducted using either HsFN3KD or HsFN3KD (D217S) mutant" here and elsewhere mutant should be variant- the organism is a mutant and the protein a variant.

Page: 16- The authors state "Asp217 H-bonds with the sugar moiety with both O3' and O4' atoms" H-bonds is not a verb, please reword.

Page: 29- The legend to Fig 1g does not explain the coloring. What is colored in cyan?

Table 1 is labelled Table S1. Also in Table 1 for an orthorhombic space group omit the angles which are set by the space group

Version 1:

Reviewer comments:

Reviewer #1

(Remarks to the Author)

The authors have appropriately responded to the comments of the referees, including mine.

Please note the following mistake in the modified text

page 24, line 29 : 'as the purified dimer completely oxidizes to monomer under reducing conditions' should be replaced by

'as the purified dimer gets completely reduced to monomer under reducing conditions'

Reviewer #2

(Remarks to the Author)

The authors have proactively addressed my comments, I am supportive of publication.

Reviewer #3

(Remarks to the Author)

My major concerns have been addressed. There are a few corrections that the authors should consider making to the revised manuscript

p. 3- "The Schiff intermediate is further hydrolyzed to separate" should read "The Schiff base intermediate..."

p. 6- "Typically, a 6 L E.coli cell expression.." should read "Typically, 6 L E. coli cell expression." And "Typically, a 3 L of Sf9" should read "Typically, 3 L of Sf9".

p. 25- "Similar electrostatic features exist in other kinases such as CLK3 and DYRK1a" References are needed here.

p. 24- "This allowed us to observe the redox state of HsFN3K with confidence, as the purified dimer completely oxidizes to monomer under reducing conditions". This is not correct- the protein does not oxidize under reducing conditions. The dimer re-equilibrates with the monomeric form under reducing conditions.

p. 17- "The α & β phosphate recognition by ..." should read "The α and β phosphate recognition by..."

If the authors think that the reason they do not see turnover in both subunits of the crystal is because of lack of Mg^{2+} ion in the active site- perhaps this explanation could be added to the manuscript and not only the response to reviewers.

Response to Reviewers

We would like to thank the reviewers for their careful review of our manuscript and for providing constructive and valuable comments. These are addressed in the manuscript with updated text and figures and additional experiments. The changes in the text are highlighted in yellow in the manuscript. The format for the references was changed according to Nature Communications style. Further explanations are provided in our point-by-point responses below.

Reviewer #1

The paper by Garg et al presents the crystal structure of fructosamine-3-kinase. This enzyme has the unique function of causing the removal of fructosamines (and other ketoamines) from proteins. The study provides not only information on the general structure of the protein, but also on the location of ATP and ADP and the fructosamine substrate in the catalytic site, allowing to identify the residues that are important for catalysis. The importance of these residues has been assessed by site directed mutagenesis and the capacity of the enzyme to cause deglycation of a ketoamine substrate has been assessed. Of particular interest is the role of a tryptophan residue (W219), which seems to limit the catalytic capacity of this enzyme.

A study on the structure of the homologous enzyme of *Arabidopsis thaliana* has previously been reported. There are some significant differences between the two structures and on the role of a conserved inter subunit disulfide bond. Furthermore, the new study provides much more information on the location of the substrate and the catalytic mechanism.

The work is well performed and the study very well described and represents a major progress in the field of deglycation and protein repair. I have the following minor comments to make.

1. In terms of substrate specificity, fructosamine 3-kinase (FN3K) differs from (human) fructosamine-related protein (FN3KRP) by its capacity to phosphorylate fructosamine (which have the phosphorylated 3' hydroxyl group in 'L' orientation), while both FN3K and FN3KRP phosphorylate ketoamine substrates (ribulosamine, psicosamine) with a 3' OH group in D orientation (Collard et al. 2003). Can this difference in the specificity be ascribed to specific residues in the catalytic site?

Thank you for raising this interesting point. Collard et al 2003 showed that FN3K is able to phosphorylate ribulosamine and fructosamine (with D and L orientation of 3' OH group respectively), while FN3K-RP is only capable of phosphorylating ribulosamine (with D orientation of 3' OH group).

Since there is no experimental structure of FN3K-RP in either the apo or in complex with the substrate, we used an AlphaFold predicted structure of human FN3K-RP and superimposed it onto our FN3K crystal structure bound with ADP and ATP to look for any differences around the sugar moiety in the substrate binding pocket. We noticed that all the residues surrounding the DMF binding site are well conserved between FN3K and FN3K-RP, except for FN3K-Asn284, which is replaced with a His284 in FN3K-RP. However, FN3K Asn284 only interacts with either the sugar 4' or 5' OH group via a water-mediated

interaction in the ADP- and ATP-bound structures, respectively, and not the 3' OH group. The FN3K-RP His284 also seems incapable of directly sensing the 3' OH in the sugar. Of course, this comparison is based on a predicted model but, unfortunately, did not provide us with enough insight to address this.

2. The choice of the peptide sequence that is studied as a substrate in figure 1A is a bit odd, since the physiological arginine in NRF2 is replaced by a lysine, to facilitate glycation. Actually, I am not sure that there is any paper in the literature showing convincingly (through a structural approach) that a ketoamines can be formed when aldoses react with an arginine. Other glycation products are formed, nonetheless. I want to stress that this is a minor point and that I am not recommending the authors to retry a more physiological substrate. The interest of the experiment is to confirm that FN3K is able to catalyze a deglycation reaction and this is convincingly shown. It may be useful to mention that the use of ribose rather than glucose as a glycating agent facilitated the demonstration of deglycation, since ribulosamine-3-phosphate (half-life \approx 20 min) is much less stable than fructosamine-3-phosphate (half-life \approx 8h).

We thank the reviewer for raising this point and for the suggestions. We agree with the reviewer that ketosamines, substrates of FN3K, can only be formed when reduced sugars react with lysines. We also acknowledge that the primary objective of this *in vitro* deglycation assay is to demonstrate that FN3K is indeed able to phosphorylate and catalyze the deglycation reaction, which has been achieved with the peptide substrate used. We have attempted to clarify these points in the revised manuscript (page 13, line 10-11). We appreciate the suggestion regarding ribulosamine-3-phosphate, which is now mentioned in the results (page 13, line 12-14).

3. The limitation of the catalytic activity of FN3K by the presence of a tryptophan at position 219 rather than a histidine may be useful to slow down side activities of this enzyme.

We agree with the reviewer that having a tryptophan in FN3K kinases might have implications in a controlled kinase activity on glycated substrates. We have now added this to the discussion (page 26, line 5-6).

Reviewer #2

The manuscript from Garg et al. describes the structural characterization of the FN3K enzyme that plays important roles in the regulation of protein glycation through direct phosphorylation of the damage to promote its spontaneous removal. The authors obtain several different crystal structures of FN3K in complex with different small molecules including a model of the glycation damage. Overall, the structures appear high quality and support the author's conclusions. I am supportive of publication by have two minor comments.

Comment #1 - The arrow pushing mechanism by where elimination of the phosphorylated hydroxyl-

group to regenerate an aldehyde at the 1-position, followed by hydrolysis of the glycation damage is not immediately obvious to me. Do the authors have a proposed mechanism? I was also unable to readily find one by searching the internet.

We thank the reviewer for raising this interesting question. Though the exact mechanism of destabilization and deglycation post-phosphorylation has not been investigated in detail so far, a mechanism that was described by Schaftingen et al 2012 (doi.org/10.1007/s00726-010-0780-3) seems plausible. This mechanism proposes that the phosphate group in C3 acts as an acid-base catalyst for the deprotonation of C1, leading to the enolization of the substrate and the β -elimination of the phosphate group forming a Schiff base, which is then hydrolyzed to restore the amino group on basic residues (see the image below). Now, we have modified the sentence in the introduction to better describe this mechanism (page 3, line 27-29).

Comment #2 - How do the authors know which glycation product they are detecting by MS is the schiff-base and which is the Amadori product? I understand that one has a different retention time, but how do they sure which is which?

We thank the reviewer for the comment. Because the Schiff base and the Amadori product have the same exact mass, they are not distinguishable by mass spectrometry. However, the Schiff base is less stable and more susceptible to fragmentation compared to its Amadori counterpart, generating through C2-C3 retro-aldol cleavage a diagnostic ion at m/z 478.29 incorporating the intact peptide and two carbon moieties from the ribose (Xing et al 2020) (doi.org/10.1016/j.carres.2020.107985). We indeed observed this signature fragment ion in the mass spectrum and have highlighted this in the new Figure S1.

Furthermore, as FN3K has been demonstrated to specifically act on various ketosamine (Szwergold et al 2001)(doi.org/10.2337/diabetes.50.9.2139) and that we didn't observe any changes to the level of the Schiff base post-incubation with FN3K in our assays (Figure 1D), we assigned the two products accordingly in our chromatogram. We have also added a sentence in the revised manuscript to elaborate on this point (Page 13, line18-20).

Reviewer #3

The manuscript by Garg et al. describes a series of crystal structures of the enzyme Homo sapiens

fructosamine-3 kinase (FN3K) in the unliganded state and in complex with nucleotide analogs and a sugar substrate mimic to identify structural determinants of substrate recognition and catalysis. The structural findings were validated for selected active-site residues by mutagenesis. The activity of the wild-type and variants were tested on a small-molecule substrate but was also tested on glycosylated lysozyme. The findings show that the enzyme is active on protein substrate and that the activity is also sugar specific. The results should be of interest to those in the areas of cell signaling, protein kinases, drug discovery, and enzyme mechanisms. The human enzyme differs from the other structurally characterized plant enzyme which increases the impact of the work. To this point, and as described below, the authors should better analyze their findings on the dimeric state of the enzyme with the literature describing a proposed regulatory role of dimerization in the Arabidopsis enzyme. There are several points where the analysis of the findings and presentation can be improved. The authors may wish to consider the following points (page numbers from pdf for review):

Page 4- The authors state "and determined a series of crystal structures of human (Hs) FN3K in the apo, and in substrate-bound forms with various nucleotides representing different catalytic-transition states." These substrates and analogs are all ground state, there are no transition-state analogs crystallized. This should be restated.

Thank you for suggesting this point. We agree and have removed the word "transition" from the sentence (Page 4, Line 29).

Page: 6- In protein preparation in methods- could some idea of protein yield be given for those trying to reproduce the work (for instance mg protein/g cell paste for bacterial cell expression).

Thank you for this suggestion. The yield of different FN3K proteins varied greatly from E.coli and from insect cells, so we have now provided a range for our protein yield from E.coli and Sf9 cells (page 6, Line 10-11 & line 27-28).

Page: 12-The authors state "We also observed a dimeric FN3K species during the purification from insect cells (Figure S1A-B), which exhibited ~60% higher kinase activity on DMF as compared to the monomeric FN3K species (Figure 1F)." It is very unusual for a monomer-dimer equilibrium to be so slow that the dimeric and monomeric species can be isolated. Upon separation, they will re-equilibrate. There is no discussion of this point and this left me very confused. Later in the manuscript it is explained that there is a disulfide bond between subunits which means that the monomer-dimer equilibrium is dependent on redox state. This needs to be at least summarized here and can be expanded upon later when the structure is described. Otherwise, it is too confusing to the reader.

We agree with the reviewer and appreciate this suggestion. As pointed out by the reviewer, we now tested if the purified HsFN3K dimer re-equilibrates to monomer via SEC under both non-reducing and reducing conditions.

Indeed, under non-reducing conditions, the dimer partially re-equilibrates to a monomer species, while it's completely converted to a monomer under reducing conditions, suggesting a redox-state-dependent HsFN3K dimer (Figure S2F-G). This is now described on page 14, line 10-14.

Though the dimeric HsFN3K has clearly higher activity than the monomeric species in the kinase assays, we cannot rule out that some dimer converts to monomer during the assay. We added this point to the results (page 15, line 13-15).

We now also verified that the redox state in HsFN3K is dependent on Cys24, since the FN3K_C24S variant does not exhibit a dimeric species anymore on size-exclusion and non-reducing SDS-PAGE analyses (Figure S2F-G). We have incorporated these results into the structure part of the results as suggested (page 15, line 5-8).

Page: 14- The authors state – “The electrostatic surface of the Apo-HsFN3K structure exhibits a negatively charged pocket near the P-loop for ATP binding, and a sulfate ion is observed in one protomer occupying the space for the nucleotide β -phosphate in this structure.” It seems odd for an ATP binding pocket to be negatively charged as ATP itself is negatively charged. Is this common in kinases with this fold? Can the authors explain this for the reader?

Thank you for raising this point. The ATP binding site in FN3K is decorated with polar residues but shows an overall negative charge, and this feature seems to have been observed in other kinases as well. We analyzed several other kinase structures and the overall charge in nucleotide binding site seems to vary substantially for different kinases. For instance, human CLK3, human DYRK1a, and bacterial MTRK have an overall negatively charged catalytic site, while human CK2, human PI3K, and bacterial HSK2 kinase have an overall positively charged catalytic site. It's possible that the divalent cation assists in nucleotide binding in the negatively charged pocket. We have mentioned this observation in the revised manuscript (Page25, line 23-26).

When I looked at the comparison with the *A. thaliana* structure and the paper describing it I note that there is a completely different interpretation of the presence of the disulfide bonded dimer- In that paper (ref 16 here) the authors state "we found that the P-loop is stabilized in an extended conformation by a Cys-mediated disulfide bond connecting two chains to form a covalently linked dimer in which the reduction of disulfides resulted in AtFN3K activation. Consistent with this, HsFN3K, in which the P-loop Cys is conserved, was redox-regulated and displayed altered oligomerization when proliferating cells are exposed to acute oxidative stress." The opposite is found here, where the monomeric form is less active. The authors should address this important difference during the discussion. It is only briefly mentioned in the discussion section. A more in-depth description of the differences in structure and the accessibility of the active site should probably be given. If space is needed I would suggest removing or shortening the long paragraph on the attempt to obtain a structure with phosphorylated-DMF, which was unsuccessful.

Thank you for raising this point. Shrestha et al 2020 showed that the purified HsFN3K protein is more active under reducing conditions, while in our hands, the HsFN3K dimer (which partly re-equilibrates to the monomer form) is clearly more active than the monomeric HsFN3K.

We think that one reason for this difference could be due to the fact that the protein purification was performed under non-reducing conditions in Shrestha et al 2020, while we performed protein purification under physiologically relevant reducing conditions. Purifying HsFN3K under non-reducing conditions might have affected the dimer-monomer equilibrium of the protein in Shrestha et al's work, as the HsFN3K dimer does not convert to the monomer even under reducing conditions, suggesting that the protein fold might have been affected during purification itself. HsFN3K purification under reducing conditions not only mimics physiological conditions better, but still allowed us to observe the redox-state dependency of HsFN3K with confidence, as the dimer completely oxidizes to monomer under reducing conditions (1mM DTT). As the reviewer suggested, we added these points in our discussion (from page 24, line 24 upto page 25 line-1).

Also, as suggested, we looked at the catalytic site accessibility between the HsFN3K and AtFN3K structures, but in both structures these sites are accessible in the respective dimers, eliminating this possibility.

Page: 15-The authors describe the differing geometries of ADP in this and the *A. thaliana* structure but do not address the importance. This is an example of a tendency for the results and discussion to be too descriptive. The authors should look to describe the mechanistic impact of the differing structures.

In members from different kinase families, a Lys or Arg residue (analogous to K41 in human FN3K) stabilizes the α and β phosphates of ATP to position them in a strictly catalysis-competent geometry in the nucleotide-binding site. The previous FN3K structure from Arabidopsis, did not show these interactions with the ADP, which likely affected nucleotide geometry. Our structures confirm that similar to other kinases, Lys41 also stabilizes the ATP α and β phosphates in FN3K. We have added a sentence to specify the importance of this observation in the revised manuscript (Page 17, line 12-15).

Page: 18- The authors state "Interestingly, a W219H mutation converts HsFN3K into a super kinase, exhibiting several-fold higher kinase activity against DMF in vitro (Figure 3E), with a significantly higher ATP turnover." As the authors kinetics show substrate saturation and they know the enzyme concentration, it seems that an analysis to yield k_{cat} and K_m is feasible for WT and mutants and would present more quantitative results.

Thank you for this suggestion. Actually, we calculated the k_{cat} and K_m values for WT-FN3K and W219H-FN3K proteins for the kinetics assays using Prism. However, we did not include them in the manuscript because the calculated k_{cat} values for the W219H were way too high, which skewed the calculation of the K_m values [$K_m = k_{off} + k_{cat} / k_{on}$]. This introduces a bias in the k_{cat} / K_m calculation as well. So, to avoid any confusion for readers, we did not add the k_{cat} or K_m values in the manuscript.

However, to be confident about the qualitative observations, we validated the effect of W219H mutation not only on the small molecule substrate DMF but also on a glycated protein substrate (Figure 6D), which shows a clear enhancement in kinase activity.

Page: 24- Obviously the enzyme can turn over in crystallo, because of the observation of ADP in the binding site of crystals prepared with ATP and substrate DMF in one subunit. Do the author have a hypothesis as to why the other subunit is poised for catalysis but does not turnover?

The FN3K_ATP-DMF structure with ATP-DMF in one subunit and ADP in the other subunit, was soaked in a mix of enzymatically prepared Phospho-DMF, ADP mix, which also had Mg^{2+} and some leftover ATP from the synthesis reaction.

Out of ~60 crystals that we soaked for different time points and solved structures, only one showed a clear electron density for both ATP and DMF, while most other crystals showed density for ATP, and poor or no electron density for DMF. Whereas, the second subunit was occupied by ADP in most crystals.

In the structure, one subunit is poised for the catalysis with both ATP and DMF bound, while could not turn over due to lack of a Mg^{2+} ion in the catalytic site. While, in most other crystals we tested the catalysis had likely occurred, explaining poor/no DMF density with ATP.

Page: 36- The names of the complexes in Methods are not consistent with those in Table I or the results section of the text. I was looking for the conditions for the ATP-DMF complex and could not find them.

We are sorry for the confusion. We have now made sure that the naming is consistent throughout the manuscript.

Typos, grammatical errors

Page: 2- "to impact the cellular function of target protein" should read "to impact the cellular function of the target protein"

Changed as suggested (Page 2, line 4).

Page 2 "of HsFN3K in apo-state," should read "of HsFN3K in the apo-state"

Changed as suggested (Page 2, line 7).

Page: 3- "Alternatively, the sugar moiety on glycated residues can be phosphorylated by a class of small-molecule kinases called fructosamine-3-kinase (FN3K), which are present in mammals, birds, and plants" The enzyme name should probably be plural to read "fructosamine-3-kinases, which are present ...". then define the abbreviation in the next paragraph - which is actually already done.

Changed as suggested (Page 3, line 21).

Page: 4- "its phosphorylation activity is believed to be crucial for its regulation on NRF2" should read ".. to be crucial for its regulation of NRF2"

Changed as suggested (page 4, line 25).

Page: 10- The authors state "all crystallization experiments were conducted using either HsFN3KD or HsFN3KD (D217S) mutant" here and elsewhere mutant should be variant- the organism is a mutant and the protein a variant.

Changed as suggested throughout the text.

Page: 16- The authors state "Asp217 H-bonds with the sugar moiety with both O3' and O4' atoms" H-bonds is not a verb, please reword.

We established the acronym for hydrogen bond in the manuscript. We also ensured that H-bond is not used as a verb in the manuscript (Page 16, line 21).

Page: 29- The legend to Fig 1g does not explain the coloring. What is colored in cyan?

Table 1 is labelled Table S1. Also in Table 1 for an orthorhombic space group omit the angles which are set by the space group

We now explained the color scheme in Fig1 legends (Page 28, line 7 & 19-20). We changed the name Table-S1 to Table-1 and removed the angles from Table-1 as well (Page 38).

Response to Reviewers

We would like to thank the reviewers for their careful review of our revised manuscript. We updated the manuscript with changes in the text highlighted in yellow. Further explanations are provided in our point-by-point responses below.

Reviewer #1 (Remarks to the Author):

The authors have appropriately responded to the comments of the referees, including mine.

Please note the following mistake in the modified text

page 24, line 29 : 'as the purified dimer completely oxidizes to monomer under reducing conditions'

should be replaced by

'as the purified dimer gets completely reduced to monomer under reducing conditions'

We changed the sentence as suggested (p17, line2).

Reviewer #2 (Remarks to the Author):

The authors have proactively addressed my comments, I am supportive of publication.

Reviewer #3 (Remarks to the Author):

My major concerns have been addressed. There are a few corrections that the authors should consider making to the revised manuscript

p. 3- "The Schiff intermediate is further hydrolyzed to separate" should read "The Schiff base intermediate..."

Changed as suggested (p3, line29).

p. 6- "Typically, a 6 L E.coli cell expression.." should read "Typically, 6 L E. coli cell expression." And "Typically, a 3 L of Sf9" should read "Typically, 3 L of Sf9".

Changed as suggested (p20, line 23).

p. 25- "Similar electrostatic features exist in other kinases such as CLK3 and DYRK1a" References are needed here.

References were added (p17, line 29-30).

p. 24- "This allowed us to observe the redox state of HsFN3K with confidence, as the purified dimer completely oxidizes to monomer under reducing conditions". This is not correct- the protein does not oxidize under reducing conditions. The dimer re-equilibrates with the monomeric form under reducing conditions.

Thanks for pointing out this mistake. Reviewer 1 also raised this. This has been corrected. (p17, line2).

p. 17- “The α & β phosphate recognition by ...” should read “The α and β phosphate recognition by...

Changed as suggested. (p9, line 15)

If the authors think that the reason they do not see turnover in both subunits of the crystal is because of lack of Mg^{2+} ion in the active site- perhaps this explanation could be added to the manuscript and not only the response to reviewers.

Thanks for this suggestion. We added a sentence about it in the results. (p10, line 27-28)